# TASTE: Text-Aligned Speech Tokenization and Embedding for Spoken Language Modeling

**Liang-Hsuan Tseng**[*23] **Yi-Chang Chen**[*1] **Kuan-Yi Lee**[23] **Da-Shan Shiu**[1] **Hung-yi Lee**[4]

[*]Equal contribution   [1]MediaTek Research   [2]Internship at MediaTek Research
[3]Graduate Institute of Communication Engineering, National Taiwan University
[4]Artificial Intelligence Center of Research Excellence, National Taiwan University
{yi-chang.chen, ds.shiu}@mtkresearch.com
{f11921067, b10901091, hungyilee}@ntu.edu.tw

## Abstract

Recent efforts target spoken language models (SLMs) that not only listen but also speak for more natural human-LLM interaction. Joint text-speech modeling is a promising direction to achieve this. However, the effectiveness of recent speech tokens for joint modeling remains underexplored. To address this, we introduce **T**ext-**A**ligned **S**peech **T**okenization and **E**mbedding (TASTE), a method that directly addresses the modality gap by aligning speech token with the corresponding text transcription during the tokenization stage. We propose a method that can achieve this through an attention-based aggregation mechanism and with speech reconstruction as the training objective. We have conducted extensive experiments to demonstrate that TASTE can preserve essential paralinguistic information while dramatically reducing the token sequence length. Moreover, TASTE enables straightforward joint spoken language modeling by using Low-Rank Adaptation on the pre-trained text LLM. Our experimental results show that joint modeling with TASTE outperforms other pre-trained SLMs in tasks such as speech continuation and likelihood-based next-speech selection, showcasing its effectiveness. To our best knowledge, TASTE is the first end-to-end approach that utilizes a reconstruction objective to learn a joint tokenization and embedding tailored for text-speech spoken language modeling. Our demo, code, and models are available at https://mtkresearch.github.io/TASTE-SpokenLM.github.io.

## 1 Introduction

Spoken language modeling (SLM) is an intriguing direction nowadays that aims at creating models that can not only listen but also *speak* (Lakhotia et al., 2021; Nguyen et al., 2023; Défossez et al., 2024; Fang et al., 2024; Arora et al., 2025). Typically, building an SLM requires two stages: first, deriving speech tokenizations; second, training a language model based on the speech tokens. For the speech tokens, previous approaches either apply self-supervised learning (SSL) representations following by discretization techniques (Baevski et al., 2020; Lakhotia et al., 2021; Nguyen et al., 2023; Hassid et al., 2023) or reuse units from neural codec models like EnCodec and SoundStream (Défossez et al., 2023; Zeghidour et al., 2021; Kumar et al., 2023; Siuzdak et al., 2024). Although autoregressive modeling with these speech tokens shows great potential in text-to-speech (TTS) (Wang et al., 2023; Xin et al., 2024a; Kim et al., 2024; Chen et al., 2024), previous SLMs that model only speech tokens (Lakhotia et al., 2021; Nguyen et al., 2023) have been shown to lack semantic fidelity (Lin et al., 2024).

To bridge this gap, one promising direction is to leverage text—which is rich in semantics—during spoken language modeling. TWIST (Hassid et al., 2023) shows that SLMs can benefit from initializing with text LLMs. Building on this idea, recent work has shifted toward joint text–speech modeling to enhance semantic coherence in generated speech. Such approaches typically adopt a hybrid decoding scheme that generates both text and speech tokens. However, combining the two modalities introduces a length mismatch, since speech token sequences are usually much longer than their textual counterparts. Common remedies include interleaving text and speech tokens (Nguyen et al., 2025) or inserting padding to synchronize sequence lengths (Défossez et al., 2024; Xie & Wu, 2024a;

Figure 1: **The concept overview.** Conventional methods extract speech tokens solely from speech, inevitably carries overlapped information with text tokens and induces length-mismatch problem when conducting joint text-speech modeling. By taking dual modalities as input, we generate speech tokenization that is aligned with text, facilitating straightforward and effective joint modeling.

Fang et al., 2024; Xie & Wu, 2024b), but these solutions rely on additional alignment procedures or heuristic rules, making joint modeling more complex.

As hybrid text–speech decoding becomes the prevailing paradigm for joint SLM (Défossez et al., 2024; Xie & Wu, 2024a; Fang et al., 2024; Li et al., 2025; Xie & Wu, 2024b), the design of speech tokens should be reconsidered in light of this setting. This motivates the development of more effective joint tokenization methods, which can be derived under the following two considerations: **1)** a speech token should avoid redundantly encoding text content—already captured by the text tokens—and instead focus on conveying paralinguistic information; and **2)** a straightforward one-to-one correspondence between text and speech tokens is preferable, as it allows SLMs to generate a text token and a speech token simultaneously without any heuristics or explicit alignments applied, mitigating the length mismatch issue during the tokenization stage.

In this work, we introduce **T**ext-**A**ligned **S**peech **T**okenization and **E**mbedding (TASTE), a special type of joint tokenization tailored for text-speech joint spoken language modeling. By acknowledging that the length mismatch introduces additional complexity in joint modeling, we develop our speech token to be aligned with its corresponding text transcription tokens. To achieve this, we first obtain the textual transcription of a speech with the ASR model; then we derive the speech token based on the transcription through a specialized cross-attention mechanism for speech reconstruction. Note that the full process can be accomplished in an end-to-end manner, with no explicit speech-text alignment required. Unlike previous speech tokens that are developed under a fixed stride with fixed down-sampling rate, our speech token has dynamic frequency as it is text-aligned. Figure 1 shows an overall concept of TASTE, illustrating how our joint tokenization allows effective joint modeling.

To evaluate the effectiveness of TASTE, we first conduct extensive experiments on speech reconstruction. Our results on LibriSpeech (Panayotov et al., 2015) show that TASTE not only resynthesizes speech in high quality, but also retains similarity to the original speech. TASTE achieves high-end reconstruction at an extremely low bit rate ($\sim$150 bps); while the other comparable methods are often more than thousands of bps. We attribute the efficiency to the involvement of text tokens during encoding and decoding, and our speech tokens focus on carrying paralinguistic information, which is backed up by the demonstration that TASTE allows simple text-aligned speech editing. By exchanging the partial text-aligned speech tokens from two different utterances with the same content, we demonstrate that the paralinguistic information such as duration and tone can be exchanged precisely following the words being exchanged, resulting in natural edited speech.

On the other hand, we demonstrate that TASTE successfully allows effective spoken language modeling. We perform straightforward joint modeling with TASTE under Low-Rank Adaptation (Hu et al., 2021). We first perform speech continuation experiments with 3-second speech prompts given. The evaluation is three-fold. We use GPT-4o for evaluating the semantic aspect; UTMOS (Saeki et al., 2022) for the acoustic aspect; and the human listening test for the general evaluation. Results show that our SLMs not only generates natural, meaningful speech continuations, but also outperforms the other 7B pre-trained SLMs across all the continuation evaluation aspects with 1.3B parameters. We also evaluate our SLMs on two benchmarks, SALMON (Maimon et al., 2024) and StoryCloze (Hassid et al., 2023) and our results show that our SLMs achieve comparable performance compared to the other text-speech joint modeling methods. Moreover, we show that our pretrained SLM can perform spoken question answering under the few-shot scenario.

In summary, we derive TASTE, a specialized tokenization approach for text–speech spoken language modeling. By aligning speech tokens with their text counterparts, TASTE provides a simple yet effective form of joint tokenization. Our results highlight joint tokenization as a key factor in joint modeling, offering a new perspective that may foster further research into more effective designs.

## 2 RELATED WORK

Recent SLMs often require speech tokenization to conduct language modeling with the next prediction objective as the text LLMs. Unlike text, the speech signal is continuous and lengthy, making it difficult to derive proper speech tokenization for spoken language modeling. Common approaches may utilize self-supervised learned (SSL) speech models followed by quantization techniques to extract speech tokens (Baevski et al., 2020; Hsu et al., 2021; Lakhotia et al., 2021; Hassid et al., 2023; Nguyen et al., 2025). In addition, audio or speech codec models have also been used for tokenization in recent SLMs (Zeghidour et al., 2021; Défossez et al., 2023; 2024; Zhang et al., 2024). These models are designed for resynthesis, where the speech decoders are jointly learned with the encoders.

With speech tokenization, GSLM (Lakhotia et al., 2021) first demonstrates the possibility of building an SLM that can generate speech. TWIST (Hassid et al., 2023) further shows that the SLM can benefit from initialization with the text-only LLM. With regard to the huge success of text-only LLMs, recent work shifts the focus towards joint speech-text modeling (Défossez et al., 2024; Xie & Wu, 2024a). To facilitate joint modeling, Spirit LM (Nguyen et al., 2025) adopts an interleaving strategy; moshi (Défossez et al., 2024) trains its own tokenizer with a reduced token frequency. Moreover, delayed or sequential generation are introduced for joint modeling (Xie & Wu, 2024a).

Despite the increasing demand of joint speech-text modeling, we do not find any work discussing the effectiveness of current speech tokenization for it. However, we have observed that recent work is trying to mitigate the modality gap by reducing the token frequency (Défossez et al., 2024; Zeng et al., 2024), conducting an additional training stage for text-speech alignment (Xie & Wu, 2024a), or leveraging hidden representations of the ASR encoder (Du et al., 2024a). This motivates us to design a speech tokenization that is directly aligned with its text counterpart, tackling the mismatch issue during the tokenization stage.

In TASTE, we utilize a specialized mechanism based on attention to aggregate the encoder representations. We clarify that the text-speech cross-attention mechanism has also been used for fine-grained control of TTS. More specifically, Chen & Rudnicky (2022) propose content-style cross-attention to indicate their text-speech cross-attention mechanism that enables style transfer in TTS. Although both utilize a specialized text-speech cross-attention mechanism, the design choices and problem formulations are completely different. We attribute of our main novelty to inventing a text-aligned speech tokenization and embedding for joint spoken language modeling, and the text-speech cross-attention mechanism is considered and shown to be a clean and effective way of achieving it.

## 3 METHOD

We propose text-aligned speech tokenization and embedding (TASTE) to facilitate effective joint speech-text spoken language modeling. Here, we first introduce how we derive our joint tokenization in Section 3.1, and then discuss how we use TASTE for spoken language modeling (§ 3.2).

### 3.1 BUILDING TASTE

As depicted in Figure 2, TASTE is comprised of the two main components: the text-aligned speech tokenizer (§ 3.1.1) that produces the text-aligned speech tokenization; and the speech decoder (§ 3.1.2) to *reconstruct* speech based on the text token and the TASTE speech token aligned with it. The training objective of speech reconstruction is described in Section 3.1.3.

### 3.1.1 TASTE SPEECH TOKENIZER

In TASTE, the speech tokenizer, denoted as $\text{Tokenizer}(\cdot)$, is designed to generate the text-aligned speech tokenization and embedding with the speech-text pair $X = (\boldsymbol{u}, \boldsymbol{v})$ taken as input, where $\boldsymbol{v}$ represents the textual transcription of the speech utterance $\boldsymbol{u}$, which can be easily obtained

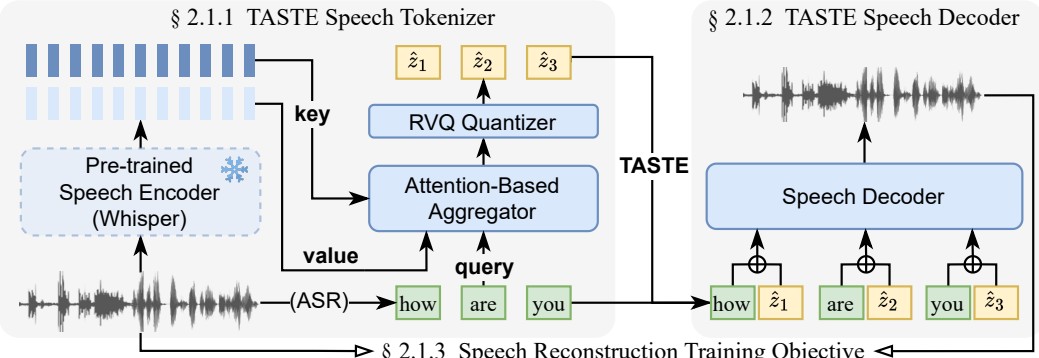

Figure 2: **The overall framework of our text-aligned speech tokenization and embedding.** The left side illustrate the process of obtaining the TASTE tokenization $\hat{z}$, detailed in Section 3.1.1; while the right side demonstrate how we reconstruct the speech with TASTE (Section 3.1.2). The training objective for our speech reconstruction is discussed in Section 3.1.3.

through an automatic speech recognition (ASR) system. Recent developments in robust and efficient ASR (Radford et al., 2023; Gandhi et al., 2023) allow us to focus on discussing how to derive the text-aligned speech token effectively by assuming that $\boldsymbol{v}$ is of sufficient quality. The TASTE speech tokenizer is composed of three major components: an *encoder*, an *aggregator*, and a *quantizer*.

The encoder $\mathrm{Encoder}(\cdot)$ contains $L$ layers of Transformer (Vaswani et al., 2017) encoder blocks and is used to extract high-dimensional speech representation. We employ the pre-trained Whisper ASR encoder (Radford et al., 2023) as our speech encoder, and it is frozen during training. For an input speech utterance $\boldsymbol{u}$, the encoder produces a sequence of hidden states from each layer $[\boldsymbol{h}^{(1)}, \boldsymbol{h}^{(2)}, \ldots, \boldsymbol{h}^{(L)}]$. In our experiments, we retain the *last* hidden representation $\boldsymbol{h}^{(L)}$ and the *shallow* representation $\boldsymbol{h}^{(l)}$ from the first half of the hidden representations of the encoder:

$$\boldsymbol{h}^{(L)}, \boldsymbol{h}^{(l)} = \mathrm{Encoder}(\boldsymbol{u}), \quad \text{where } 1 \le l \le \left\lfloor \frac{L}{2} \right\rfloor.$$

Note that both of the hidden representations $\boldsymbol{h}^{(L)}, \boldsymbol{h}^{(l)} \in \mathbb{R}^{T \times d_h}$ have their length denoted as $T$ and the hidden dimension indicated by $d_h$.

The hidden representations extracted from the encoder are then passed to the *aggregator*. The aggregator is designed to obtain a more compressed speech representation $\boldsymbol{z}$ that is aligned in length with the text transcription $\boldsymbol{v}$. Consider that $\boldsymbol{v} = [v_1, v_2, \ldots, v_N], v_i \in \mathbb{V}$ is a text token sequence with length $N$, the input and output of the aggregator can be denoted as:

$$\boldsymbol{z} = \mathrm{Aggregator}(\boldsymbol{v}, \boldsymbol{h}^{(L)}, \boldsymbol{h}^{(l)}), \text{ where } \boldsymbol{z} \in \mathbb{R}^{N \times d_z}, \boldsymbol{v} \in \mathbb{V}^N, \text{ and } \boldsymbol{h}^{(L)}, \boldsymbol{h}^{(l)} \in \mathbb{R}^{T \times d_h}.$$

To make the speech representation $\boldsymbol{z}$ text-aligned, we conduct a simple yet effective attention mechanism based on the three inputs. Consider that the original multi-head attention in Vaswani et al. (2017) is denoted as $\mathrm{MultiHead}(Q, K, V)$, our first layer attention in the aggregator takes:

$$Q = \text{text transcription } \boldsymbol{v}, \quad K = \text{encoder last hidden } \boldsymbol{h}^{(L)}, \quad V = \text{encoder shallow hidden } \boldsymbol{h}^{(l)}.$$

By doing so, the length of our first multi-head attention output should follow the text transcription $\boldsymbol{v}$. Note that the query of the following layers becomes the output from the previous layer. In addition, intuitions of using the encoder's last hidden representation as keys, and the shallow hidden representation as values can be described as follows: **1)** In Transformer-based ASR models, the last hidden states often encode rich speech-text alignment cues; sometimes the cross-attention weight matrices can even be exploited as soft word-alignment maps (Radford et al., 2023; Gandhi et al., 2023). **2)** The shallow representation has been shown to support high-quality speech reconstruction even when the quantization is applied (Du et al., 2024a;b). Based on the above observations, we design our aggregator that can use the soft attention maps obtained from the last encoder representations and the text transcriptions, to aggregate the shallow encoder representations that have been shown to be beneficial for high-end speech reconstruction.

After getting the text-aligned representation, the quantizer $\mathrm{Quantizer}(\cdot)$ is adopted to discretize the text-aligned representation. We use the residual vector quantization (RVQ) to allow coarse-to-fine quantization. Given the text-aligned speech representation $\boldsymbol{z}$ and the quantizer containing $R$ residual vector quantization layers, we generate:

$$\boldsymbol{q}, \hat{\boldsymbol{z}} = \mathrm{Quantizer}(\boldsymbol{z}), \qquad \boldsymbol{q} = [\boldsymbol{q}^{(1)}, \boldsymbol{q}^{(2)}, \ldots, \boldsymbol{q}^{(R)}], \quad \hat{\boldsymbol{z}} = \sum_{r=1}^{R} \hat{\boldsymbol{z}}^{(r)} \tag{1}$$

where each $\boldsymbol{q}^{(r)} \in \mathbb{C}^N$ denotes the $r$-th layer code sequence with code set $\mathbb{C}$; and the quantized embedding $\hat{\boldsymbol{z}}$ to be the summation over each layer of the codebook vectors. Note that both of the code sequence and the quantized speech embedding $\hat{\boldsymbol{z}}$ are text-aligned, with the lengths to be $N$.

### 3.1.2 TASTE Speech Decoder

The speech decoder aims to perform speech reconstruction conditioned on the text token sequence and the text-aligned speech tokenization. As shown in Figure 2, the text and speech tokens are aligned in lengths and being fed into the speech decoder after weighted sum in an autoregressive manner. The speech decoder is composed of the two components: the unit decoder and the unit-to-speech vocoder.

The unit decoder $\mathrm{UnitDecoder}(\cdot)$ is a Transformer-based decoder that takes the text token sequence $\boldsymbol{v}$ and the aligned speech embedding $\hat{\boldsymbol{z}}$ as condition and predicts the speech unit $\boldsymbol{y}$ for reconstruction:

$$\boldsymbol{y} = \mathrm{UnitDecoder}(\hat{\boldsymbol{z}}, \boldsymbol{v}). \tag{2}$$

Note that the additional speaker embedding is also taken as input to facilitate global speaker voice control in our spoken language models (Ju et al., 2024). After we generating the speech unit $\boldsymbol{y}$, we use a unit-to-speech vocoder to further transform the unit into the reconstructed speech.

### 3.1.3 Training Objective

Similar to other reconstruction-based speech tokens (Zhang et al., 2024; Liu et al., 2025), we derive TASTE by training it for speech resynthesis. To achieve this, we extract the speech unit $\boldsymbol{y}^{\mathrm{target}}$ with length $T'$ from the original speech $u$ as the target unit for our speech tokenizer and speech decoder. Given the text transcription $\boldsymbol{v}$, the TASTE speech embedding $\hat{\boldsymbol{z}}$, and the unit from the original speech $\boldsymbol{y}^{\mathrm{target}}$ as the target, the speech reconstruction through the tokenizer and the unit decoder parametrized by $\theta$ under the next prediction schema can be considered as minimizing the cross-entropy loss below:

$$\mathcal{L}_{\mathrm{ce}}(\theta) = \frac{1}{|T'|} \sum_{t=1}^{T'} -\log p_\theta(y_t^{\mathrm{target}} | \hat{\boldsymbol{z}}, \boldsymbol{v}; \boldsymbol{y}_{<t}^{\mathrm{target}}) \tag{3}$$

On the other hand, we employ the quantization loss as well to tokenize the continuous representation $\boldsymbol{z}$ extracted from the encoder-aggregator. Following prior works (Défossez et al., 2023; Zeghidour et al., 2021), given that $\boldsymbol{z}^{(r)}$ is the $r$-th residual and $\hat{\boldsymbol{z}}^{(r)}$ indicates the $r$-th quantized residual, the the commitment loss is defined as:

$$\mathcal{L}_{\mathrm{rvq}}(\theta) = \sum_{r=1}^{R} \| \boldsymbol{z}^{(r)} - \hat{\boldsymbol{z}}^{(r)} \|. \tag{4}$$

By summation over both losses, we formulate the overall loss for training TASTE as:

$$\mathcal{L}_{\mathrm{taste}} = \mathcal{L}_{\mathrm{ce}} + \mathcal{L}_{\mathrm{rvq}}. \tag{5}$$

## 3.2 TASTE for Spoken Language Modeling

Next, we describe how we conduct effective spoken language modeling with TASTE. Following previous work (Hassid et al., 2023; Nguyen et al., 2025), we perform pre-training on speech data. The text transcription of the speech data is also used for joint speech-text pre-training of our text-aligned spoken language model (TASLM). Since TASTE tokenization already aligns with the text token sequence, we can conduct a straightforward joint modeling, as illustrated in Figure 1. To demonstrate the robustness of TASTE, we perform two types of text-aligned spoken language modeling. First, we build $\mathrm{TASLM}_{\mathrm{token}}$ over our text-aligned speech **token** $\boldsymbol{q}$, discussed in Section 3.2.1. Then, we show how we build $\mathrm{TASLM}_{\mathrm{emb}}$ with our text-aligned speech **embedding** $\hat{\boldsymbol{z}}$, detailed in Section 3.2.2.

### 3.2.1 MODELING TASTE TOKEN

As our speech tokens derived from the RVQ quantizer contain $R$ layers of codes, we employ $R$ linear heads for multi-head prediction in our $\text{TASLM}_{\text{token}}$. Namely, the $\text{TASLM}_{\text{token}}$ simultaneously predicts the next text token and the corresponding $R$ layers of speech tokens in each step. The overall training objective follows the original next token prediction scheme, but with multiple predictions across modalities at each step. Specifically, given the text transcription $\boldsymbol{v}$ and $R$ layers of quantized RVQ codes $\boldsymbol{q}$, the multi-head next-token prediction training objective can be formulated as:

$$\mathcal{L}_{\text{token}}(\phi) = \frac{1}{|N|} \sum_{i=1}^{N} \Big( -\log p_\phi^{\text{text}}\big(v_i\big|\boldsymbol{v}_{<i}, \boldsymbol{q}_{<i}\big) + \sum_{r=1}^{R} -\log p_\phi^{(\text{r})}\big(q_i^{(\text{r})}\big|\boldsymbol{v}_{<i}, \boldsymbol{q}_{<i}\big) \Big), \qquad (6)$$

with $\phi$ represents the parameter of the $\text{TASLM}_{\text{token}}$, and $p^{(r)}$ is the $r$-th probability prediction for the $r$-th RVQ code. As for inference, we directly sample the codes and the text simultaneously, and transform the codes into the corresponding embedding for the speech decoder to generate speech.

### 3.2.2 MODELING TASTE EMBEDDING

Besides the token code sets, recent progress on latent modeling (Kim et al., 2024; Meng et al., 2024; Sun et al., 2024; Fan et al., 2025) motivates us to conduct experiments on modeling our text-aligned speech embedding. Referencing MELLE (Meng et al., 2024), we employ a linear layer that predicts the mean vector $\mu_i$ and a log-magnitude variance vector $\log \sigma_i^2$, where $i$ indicates the $i$-th frame of the sequence. And the final predicted latent of frame $i$ is denoted as $e_i = \mu_i + \sigma_i \odot \epsilon$, where $\epsilon \sim \mathcal{N}(0, I)$. Following MELLE, the straight-through estimator is applied to allow gradients to back-propagate properly during training.

To facilitate latent prediction, we apply the regularization loss and the Kullback-Leibler (KL) divergence loss druing training, which is described as follows:

$$\mathcal{L}_{\text{reg}}(\psi) = \|e_\psi - \hat{\boldsymbol{z}}\|_2^2, \quad \mathcal{L}_{\text{KL}} = \frac{1}{2} \sum_{i=1}^{N} \sum_{j=1}^{d_z} \big(\sigma_i[j] + (\mu_i[j] - \hat{z}_i[j])^2\big) - 1 - \log \sigma_i^2[j]\big), \quad (7)$$

where $\psi$ indicates the parameter of $\text{TASLM}_{\text{emb}}$, and $d_z$ is the dimension of our text-aligned embedding $\hat{\boldsymbol{z}}$. The regularization loss $\mathcal{L}_{\text{reg}}$ is adopted to predict close latent towards the target embedding $\hat{\boldsymbol{z}}$. The KL divergence loss calculates the KL divergence between the predicted latent distribution and the target distribution. Following MELLE, we select the target distribution to be $\mathcal{N}(\hat{z}_i, I)$. This allows simplification of $\mathcal{L}_{\text{KL}}$, which can then be approximated with the predicted vectors $\mu_i, \sigma_i$, and the target embedding $\hat{z}_i$. Finally, the overall loss along with the text loss is described as:

$$\mathcal{L}_{\text{emb}}(\psi) = \lambda_{\text{reg}} \cdot \mathcal{L}_{\text{reg}} + \lambda_{\text{KL}} \cdot \mathcal{L}_{\text{KL}} + \frac{1}{|N|} \sum_{i=1}^{N} -\log p_\psi^{\text{text}}\big(v_i\big|\boldsymbol{v}_{<i}, \hat{\boldsymbol{z}}_{<i}\big), \qquad (8)$$

where $\lambda_{\text{reg}}, \lambda_{\text{KL}}$ to be the weighted coefficients of the two losses, respectively.

## 4 EXPERIMENT SETUP

**Model Configuration** For our TASTE speech tokenizer, we initialize our encoder from Whisper (Radford et al., 2023). By doing so, we can reduce computational cost between obtaining the ASR transcription and extracting the TASTE tokenization with the TASTE encoder frozen during training. On the other hand, we use the S3 token from Du et al. (2024a) as the target unit for speech reconstruction. Since their speech tokenization facilitates additional speaker embedding, we follow the same procedure to obtain one. Adding speaker embedding allows global speaker voice control, which is a reasonable and useful scenario for spoken language models. The unit-to-speech vocoder is comprised of a flow model (Lipman et al., 2022; Mehta et al., 2022) and a HifiGAN. We use the published pre-trained ones from Du et al. (2024a), and they are not involved in our training. For the quantizer, we set the RVQ layer $R = 4$, the codebook size $512$, and the codebook dimension to be $256$. For the spoken language modeling, we follow previous work (Hassid et al., 2023) and initialize our spoken language model from a text LLM. However, this introduces the vocabulary mismatch problem between the ASR and LLM. We resolve this issue by using word-level TASTE tokenization and embedding, which is detailed in Appendix A.4. Moreover, we conduct LoRA fine-tuning of our TASLMs, with hyperparameters rank $r = 64$ and $\alpha = 128$.

Table 1: **The speech tokenization evaluation results** on the *test-clean* split of LibriSpeech. The evaluation is separated into the QUALITY and the SIMILARITY assessments, as introduced in Section 5.1.1. We use gray text to indicate the worst-performing methods in each metric. Freq. indicates the number of tokens per second. All reported results already account for the effect of ASR errors whenever textual transcriptions are involved (Text-only and TASTE).

| Method | Freq. | Bitrate | QUALITY | | | SIMILARITY | | | |
|---|---|---|---|---|---|---|---|---|---|
| | | | WER ↓ | UTMOS | DNSMOS | ViSQOL | Drtn. Con. | Spkr. Sim. | MUSHRA |
| Ground Truth | 16k | 256k | 2.1% | 4.09 | 3.84 | - | - | - | 76.6 |
| Encodec$^\alpha$ | 75 | 1500 | 5.1% | 1.58 | 3.26 | 3.46 | 0.94 | 0.63 | - |
| | 75 | 3000 | 2.6% | 2.35 | 3.48 | 3.81 | 0.96 | 0.78 | 25.6 |
| SpeechTokenizer$^\beta$ | 50 | 500 | 5.2% | 1.27 | 2.99 | 2.80 | 0.94 | 0.35 | - |
| | 50 | 2000 | 3.0% | 3.56 | 3.60 | 3.65 | 0.97 | 0.80 | 53.9 |
| | 50 | 4000 | 2.5% | 3.90 | 3.76 | 4.03 | 0.98 | 0.92 | - |
| Mimi$^\gamma$ | 12.5 | 1000 | 3.1% | 3.60 | 3.60 | 3.62 | 0.96 | 0.82 | 67.6 |
| S3 token$^\theta$ (topline) | 25 | 600 | 3.0% | 4.18 | 3.90 | 3.30 | 0.96 | 0.82 | 70.2 |
| Text-only (baseline) | ∼3 | ∼50 | 5.9% | **4.31** | **4.11** | 2.44 | 0.57 | 0.78 | 42.6 |
| TASTE (ours) | ∼**3** | ∼**150** | 4.4% | 4.29 | 4.10 | 3.05 | 0.91 | 0.80 | 68.3 |

$^\alpha$ Défossez et al. (2023), $^\beta$ Zhang et al. (2024), $^\gamma$ Défossez et al. (2024), $^\theta$ Du et al. (2024a)

**Dataset** We use two datasets–*Emilia* and *LibriTTS*–as our training datasets. Emilia (He et al., 2024) is an in-the-wild dataset where the speech is web-scaled and the transcriptions are pseudo-labeled. We use only the English subset of this multi-lingual corpus, which is about 40,000 hours. LibriTTS (Zen et al., 2019) is a reading-style corpus based on LibriSpeech (Panayotov et al., 2015). We use all the training splits in LibriTTS for training, which is approximately 600 hours of speech. In addition, the *test-clean* split in LibriSpeech is used for evaluation purposes for our TASTE tokenizer and TASLMs.

## 5 RESULT

We separate our experimental results into two parts. Section 5.1 discusses how TASTE strikes a good reconstruction quality while enables effective joint spoken language modeling; while Seciton 5.2 presents the additional results and ablation study of our joint tokenization and text-aligned SLM.

### 5.1 MAIN RESULTS

To demonstrate the benefits of our joint tokenization, we first evaluate the performance of TASTE on speech reconstruction; then introduce how it allows effective spoken language modeling. For simplicity, the evaluation metrics are introduced within each section.

#### 5.1.1 TASTE FOR SPEECH RECONSTRUCTION

**Evaluation** We evaluate our joint tokenization on two aspects: QUALITY and SIMILARITY. For QUALITY assessment, we use ASR-WER, UTMOS (Saeki et al., 2022), and DNS-MOS (Reddy et al., 2021) as our metrics for evaluation. In ASR-WER, we use HuBERT-Large (Hsu et al., 2021) as the ASR model to transcribe the reconstructed speech, and then calculate the word-error rate (WER) on the transcription. [1] UTMOS and DNS-MOS are both neural-based MOS predictors. While both evaluate the speech quality, the design purpose of DNS-MOS makes it more suitable for evaluation regarding the noise levels. For SIMILARITY assessment, we measure ViSQOL, duration consistency (Drtn. Con.), speaker similarity (Spkr. Sim.), and the MUSHRA human listening test score. ViSQOL (Chinen et al., 2020)is a production-ready tool that predicts speech quality via spectro-temporal image similarity comparisons. For the duration consistency, we first get the word-level alignment of the transcriptions of the original and the reconstructed speech using Montreal Forced Aligner (McAuliffe et al., 2017); then we calculate if the duration between each of the same words is matched under a preset tolerance window, which is set to 50 milliseconds. In the MUSHRA human listening test, we follow the original protocal (Series, 2014) to instruct evaluators to rate similarity and quality on a scale of 1 to 100 with reference given.

---

[1]https://huggingface.co/facebook/hubert-large-ls960-ft

**Results Analysis**   Table 1 reports the results of speech reconstruction on LibriSpeech. To better understand the effectiveness of TASTE, we highlight three main observations. **1)** Since our tokens are text-aligned, TASTE operates at the lowest frequency and bitrate among all tokenization methods. We estimate these dynamic values by counting the total number of tokens and accumulating the duration over the testing set. **2)** Despite this extremely low bitrate, TASTE achieves on-par or even superior performance to higher-bitrate methods in the quality assessment. In particular, TASTE yields lower ASR-WER than the text-only baseline, which we attribute to speech tokens carrying paralinguistic information that improves the naturalness of reconstructed speech. **3)** In terms of similarity, TASTE performs comparably to high-bitrate, fixed down-sampling methods across multiple metrics. The inferior results on ViSQOL can be partly attributed to our use of a flow-based vocoder, as both TASTE and the S3 token topline exhibit weaker ViSQOL performance—a phenomenon also observed in Liu et al. (2025). This degradation on ViSQOL is not reflected in the MUSHRA listening test, where TASTE attains competitive perceptual quality and similarity from a human perspective. In general, TASTE significantly outperforms the text-only baseline, confirming that it carries sufficient paralinguistic information to allow high-quality speech reconstruction.

### 5.1.2   TASTE FOR SPOKEN LANGUAGE MODELING

TASTE is designed specifically to enable effective joint spoken language modeling (SLM). To examine its effectiveness, we train pretrained SLMs on top of TASTE following the methodology in Section 3.2. In line with prior work (Nguyen et al., 2025; Lin et al., 2024), we evaluate these models from two perspectives: speech continuation evaluation and likelihood-based evaluation.

**Speech Continuation Evaluation**   First, each pretrained SLM is conditioned on 3-second speech segments from LibriSpeech *test-clean* to generate speech continuations under their own decoding schemes, following Hassid et al. (2023); Lin et al. (2024). The generated continuations are then evaluated along two main aspects: *semantic coherence* and *speech naturalness*. For the semantic aspect, we transcribe the continuations using ASR and ask GPT-4o to assign MOS scores based on their coherence. For the speech naturalness aspect, we compute UTMOS as an objective score of speech quality. In addition, human evaluators provide an overall MOS score that jointly considers both coherence and naturalness. The detailed instructions given to GPT-4o and human evaluators are provided in Appendix A.3.2.

**Likelihood-Based Evaluation**   Following previous work (Hassid et al., 2023; Nguyen et al., 2025; Lin et al., 2024), we also evaluate the SLMs through likelihood-based benchmarks, where the accuracy score is based on whether the pretrained SLM chooses the correct continuation from the two given speech utterances based on its output likelihoods. We adopt two established benchmarks SALMON (Maimon et al., 2024) and spoken StoryCloze (Hassid et al., 2023; Mostafazadeh et al.,

Table 2: **Pretrained SLM speech continuation and likelihood-based next-speech selection results.** The superscripts at the bottom of the table indicate the base models used by each SLM, indicated by superscripts. Cascade models refer to the pipeline with ASR (Radford et al., 2023), text continuation by LMs (Touvron et al., 2023), and TTS (Du et al., 2024a). This allow us to evaluate SLMs with cascade models in continuation perspective.

| Method | Finetuned / base parameters | CONTINUATION | | | LIKELIHOOD | | |
|---|---|---|---|---|---|---|---|
| | | GPT-4o | UTMOS | Human | SALMON | StoryCloze | Overall |
| *Cascade* | | | | | | | |
| Cascade (LLaMA3.2-1B$^\alpha$) | - | 3.15 | **4.25** | **4.00** | - | - | - |
| Cascade (LLaMA2-7B$^\beta$) | - | **3.43** | **4.25** | 3.98 | - | - | - |
| *Spoken LMs* | | | | | | | |
| TWIST 1.3B (Hassid et al., 2023) | 1.3B / 1.3B$^\theta$ | 1.48 | 3.25 | 1.95 | 62.5 | 61.5 | 62.0 |
| TWIST 7B (Hassid et al., 2023) | 7B / 7B$^\gamma$ | 1.44 | 3.27 | 2.04 | 63.4 | 64.7 | 64.1 |
| Spirit LM (Nguyen et al., 2025) | 7B / 7B$^\beta$ | 2.79 | 3.41 | 2.38 | 59.1 | 72.0 | 65.6 |
| Spirit LM Expr. (Nguyen et al., 2025) | 7B / 7B$^\beta$ | 1.90 | 3.40 | 2.41 | **69.0** | 66.2 | 67.6 |
| Baseline (S3 token) | 45M / 1.3B$^\alpha$ | 1.37 | 4.04 | 2.84 | 50.2 | 58.7 | 54.5 |
| TASLM 1B (token) | 45M / 1.3B$^\alpha$ | 3.08 | 4.07 | 3.93 | 60.8 | 76.5 | **68.7** |
| TASLM 1B (embed.) | 45M / 1.3B$^\alpha$ | **3.16** | **4.22** | **4.16** | 57.7 | **76.7** | 67.2 |

Base models: $^\alpha$LLaMA3.2-1B, $^\beta$LLaMA2-7B, $^\gamma$LLaMA-7B, $^\theta$OPT-1.3B

2016), which covers the acoustic aspect and the semantic aspect, respectively. Since both benchmarks contain multiple tasks, we report the average accuracy across these tasks within each benchmark for simplicity. The detailed results are in Appendix A.1.5 for the interested readers. We also report the mean of the SALMON and StoryCloze as an overall assessment for both aspects.

**Results Analysis** The results of TASLM compared to other pre-trained SLMs are shown in Table 2, and three main advantages can be observed. **1)** Compared to other pretrained SLMs, TASLM achieves substantially better performance on speech continuation across both human and machine evaluations, while also performing competitively on the likelihood-based benchmarks. Notably, this is achieved with only LoRA finetuning on a relatively small 1.3B base language model, illustrating the effectiveness of TASTE for joint modeling. **2)** Compared to cascade models with the same base LM, our $\text{TASLM}_{\text{emb}}$ achieves comparable scores on GPT-4o but higher human MOS. This indicates that its generated speech is more natural than cascade systems that rely solely on TTS during continuation. TASLM is the only SLM that not only maintains but even surpasses the performance of its corresponding text-based model, highlighting the importance of speech token modeling. **3)** Directly using the S3 token for joint modeling following Xie & Wu (2024a) yields poor performance across *all* aspects, even though it surpasses TASTE in speech reconstruction. This shows that while reconstruction quality is critical, it is not the sole consideration in tokenization for spoken language modeling. Taken together, these results highlight the central contribution of TASTE: **building a joint tokenization that facilitates more effective joint spoken language modeling**.

## 5.2 ADDITIONAL RESULTS

### 5.2.1 TASTE FOR TEXT-ALIGNED SPEECH EDITING

Beyond the main results presented above, we report several intriguing observations that further showcase the versatility of TASTE. The first is that TASTE naturally enables *text-aligned speech editing*, as illustrated in Figure 3. Suppose we have two utterances with the same transcript but different paralinguistic characteristics. By exchanging their TASTE token sequences word by word, we ask whether the associated paralinguistic traits are transferred as well. To make the effect clear, we select utterances that differ mainly in speaking rate and examine duration changes using MFA (McAuliffe et al., 2017). As illustrated in Figure 3, swapping tokens at specific word positions causes the corresponding words to exhibit clear duration shifts, while untouched words preserve their original timing—evidence that TASTE enables precise, text-aligned manipulation. This observation also echoes our design principle introduced in Section 1: a speech token should avoid redundantly encoding text content and instead concentrate on conveying paralinguistic information. Additional examples targeting other paralinguistic dimensions are provided on our demo page.

### 5.2.2 TASLM FOR SPOKEN QUESTION ANSWERING

Next, we intriguingly find out that our TASLM exhibits spoken QA ability under few-shot scenario (Brown et al., 2020). We are the only pretrained SLM in Table 2 that exhibits this capability. As

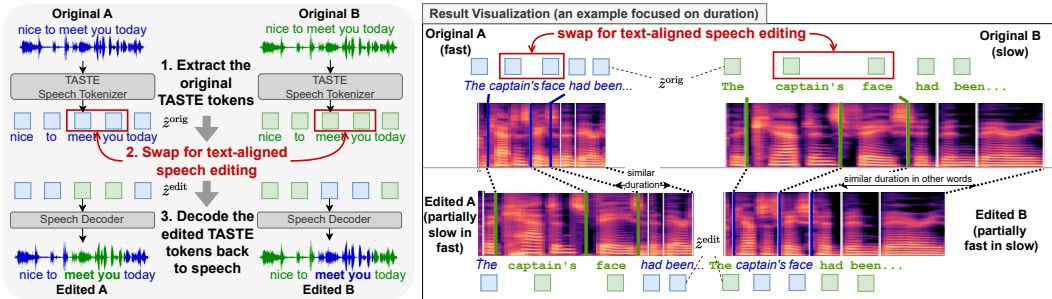

Figure 3: **An illustration of TASTE for text-aligned speech editing.** On the left shows the process of our text-aligned speech editing. We first extract the TASTE tokens; swap the tokens partially; and then decode the edited TASTE tokens into edited speech. On the right shows an example visualization. Only the durations of the words with exchanged TASTE tokens show significant difference.

Table 3: **Evaluation of spoken question answering.** Performance across modalities is compared row-wise, where T is text and S denotes speech.

| Method | Mode | Web Q. | LLaMA-Q. |
|---|---|---|---|
| Mini-Omni 0.5B(T→T) | T | 21.3 | 39.0 |
| Mini-Omni 0.5B | T+S | 4.5 | 11.6 |
| Helium 7B (text) | T | 32.3 | 75.0 |
| Moshi 7B | T+S | 26.6 | 62.3 |
| LLaMA3.1-8B-Instruct | T | 60.4 | 71.7 |
| Llama-Omni-8B | T+S | 35.5 | 67.3 |
| LLaMA3.2-1B[†] | T | 24.0 | 51.0 |
| TASLM 1B (embed.)[†] | T+S | 27.1 | 57.6 |

[†] We apply few-shot learning to facilitate question answering.

Table 4: **Ablation study on the effects of each module in TASTE speech tokenizer.** Enc. is *encoder*, Agg. is *aggregator*, and Quan. is *quantizer*. *: top-5 accuracy.

| Modules | Freq. | S3 token Acc.* |
|---|---|---|
| Enc. | 50Hz | 0.98 |
| Enc. + Agg. | ∼3Hz | 0.88 |
| Enc. + Agg. + Quan. | ∼3Hz | 0.76 |
| Enc. (*last*) | 50Hz | 0.84 |
| Enc. + Agg. (*last*) | ∼3Hz | 0.78 |
| Text-only | ∼3Hz | 0.65 |

a result, we compare it against other instruction-finetuned joint SLMs in Table 3 to better understand the performance. We use two spoken question answering benchmarks, Web Questions (Berant et al., 2013) and LLaMA-Questions (Nachmani et al., 2024), following Défossez et al. (2024). We report the accuracy of answer containment. To more comprehensively assess the impact of adding speech, we also report the performance of each system's underlying base text LLM. Notably, TASLM is the only approach that preserves its base text LLM's performance. We attribute this to TASTE's joint tokenization strategy. Specifically, we employ a straightforward one-to-one mapping between text and speech tokens, which enables simple and effective joint modeling.

### 5.2.3 ABLATION STUDY ON TASTE SPEECH TOKENIZER

We run an ablation on TASTE speech tokenizer and use S3 token top-5 reconstruction accuracy as a proxy for information retention. Table 4 first covers the module-wise ablations of our *encoder*, *aggregator*, and *quantizer*. The *aggregator* sharply reduces token rate with only a small drop in accuracy. Adding the *quantizer* lowers accuracy further, but performance is still well above the text-only baseline. Secondly, we show that using only the *last* hidden state $\mathbf{h}^{(L)}$ performs worse than using the *shallow* hidden states $\mathbf{h}^{(l)}$ (as values for the aggregator), confirming our design choice.

## 6 CONCLUSION

In this work, we propose Text-Aligned Speech Tokenization and Embedding (TASTE), to facilitate joint text-speech spoken language modeling. By aggregating proper encoder representation through the specialized cross-attention mechanism and taking the ASR model as initialization, we make the speech tokenization text-aligned in an end-to-end manner with no explicit word alignment required. With our text-aligned speech tokenization and embedding, joint text-speech modeling becomes straightforward and effective. We conduct extensive experiments demonstrating the benefits of developing a joint tokenization tailored for spoken language modeling. We anticipate that these findings encourage further research on more effective joint tokenization for generative modeling.

**Limitation** Several limitations of our current work suggest promising avenues for future development. First, while our pretrained spoken language model generates high-quality audio continuations, it lacks mechanisms for turn-taking and instruction following; developing a dialogue system is a practical next step. Second, TASTE has so far been evaluated on English; confirming its generalizability across other languages remains future work. Third, our tokenization method is tailored for joint SLMs, and its applicability to other generative tasks remains underexplored. Fourth, our pipeline currently focuses on single-speaker speech with lexical content and does not explicitly handle multi-speaker, overlapping, or non-lexical events (*e.g.*, laughter, coughing). Future work could support these capabilities by incorporating target speech extraction (Zmolikova et al., 2023) and non-lexical event tags. Finally, system latency and streaming performance are yet to be optimized for real-time applications. Overall, none of these limitations is a fundamental barrier; rather, they are natural extensions and research targets that will further enhance the versatility of TASTE framework.

**Ethics Statement** TASTE enables the efficient development of spoken language models. It lowers the barrier to building speech systems and improves the accessibility and convenience of human–computer interaction. At the same time, it raises security concerns: systems built with TASTE can more easily mimic a person's voice and synthesize convincing personalized speech. Moreover, TASTE's text-aligned speech editing makes voice manipulation straightforward. Overall, TASTE offers clear utility for beneficial applications, but responsible deployment—paired with consent, provenance, and anti-abuse safeguards—is essential to mitigate misuse risks. On the other hand, this study includes human evaluations in the form of subjective listening tests. Annotators were recruited from Amazon Mechanical Turk and compensated fairly according to the platform's recommended rates. All participants provided informed consent, and no personal information has been collected. The audio material used in the evaluation does not contain sensitive content. The study adheres to the ethical standards commonly adopted in speech perception research.

## ACKNOWLEDGEMENT

This work is partially supported by the National Science and Technology Council (NSTC) in Taiwan under Grant NSTC 113-2634-F-002-008, and Liang-Hsuan Tseng is supported by the NSTC Graduate Research Fellowship (NSTC-GRF). We also extend our gratitude to the anonymous reviewers for their constructive comments, which can be found at https://openreview.net/forum?id=6STb8DauN1.

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

Table 5: The ablation study on how the ASR affects the performance of our TASTE tokenizer regarding speech reconstruction. GT: ground-truth transcription.

| Method | WER | UTMOS | DNS-MOS | ViSQOL | Drtn. Con. | Spkr. Sim. |
|---|---|---|---|---|---|---|
| TASTE (w/ ASR) | 4.4% | 4.29 | 4.10 | 3.05 | 0.91 | 0.80 |
| TASTE (w/ GT) | 4.6% | 4.24 | 4.08 | 3.06 | 0.91 | 0.81 |

Table 6: The ablation study on how the ASR affects our SLM on spoken QA.

| Methods | Web-Q | LLaMA-Q |
|---|---|---|
| TASLM (w/ ASR) | 27.1 | 57.6 |
| TASLM (w/ GT) | 28.0 | 57.7 |

Table 7: The ablation study on using a different ASR model regarding the SLM continuation semantic evaluation. Overall, we do not observe significant *relative* performance difference.

| Evaluation Models | TWIST 1.3B | TWIST 7B | Spirit LM | Spirit LM Expr. | S3 token | TASLM (token) | TASLM (embed.) |
|---|---|---|---|---|---|---|---|
| Whisper + GPT-4o | 1.48 | 1.44 | 2.79 | 1.90 | 1.37 | **3.08** | **3.16** |
| nvidia-parakeet + GPT-4o | 1.38 | 1.49 | 2.76 | 2.06 | 1.42 | **3.20** | **3.37** |

# A APPENDIX

## A.1 SUPPLEMENTARY RESULTS

### A.1.1 ABLATION STUDY ON THE EFFECT OF ASR

Because our tokenization, SLM, as well as the evaluation using GPT-4o all rely on an ASR system to extract text transcriptions, we conduct several ablation studies to assess the impact of ASR on performance. **1)** In Table 5 and Table 6, we study how the ASR affects the performance of our TASTE tokenizer on speech reconstruction and TASLM on spoken question answering. Our results indicate that on both the tokenization and the SLM stages, the performance drop introduced by the ASR errors are almost negligible, primarily attributed to the robustness of recent ASR systems. Note that we do not use any ground-truth transcriptions in the previous experiments in the main text. **2)** We study how substituting the ASR used to produce transcripts before GPT-4o's semantic-coherence evaluation affects the reported scores. As shown in Table 7, we use another ASR model named nvidia-parakeet (Sekoyan et al., 2025), which employs an RNN-T (Graves, 2012; Xu et al., 2023) backbone. The results indicate that there is no significant relative performance difference between using Whisper and nvidia-parakeet ASR systems. TASLMs achieve much better results in both evaluation setups compared to the other pretrained SLMs.

### A.1.2 AGGREGATOR CROSS ATTENTION VISUALIZATION

To understand whether the aggregator has learned the text-speech aligned pattern, we visualize its cross attention map across all the layers and heads in Figure 4 and Figure 5.

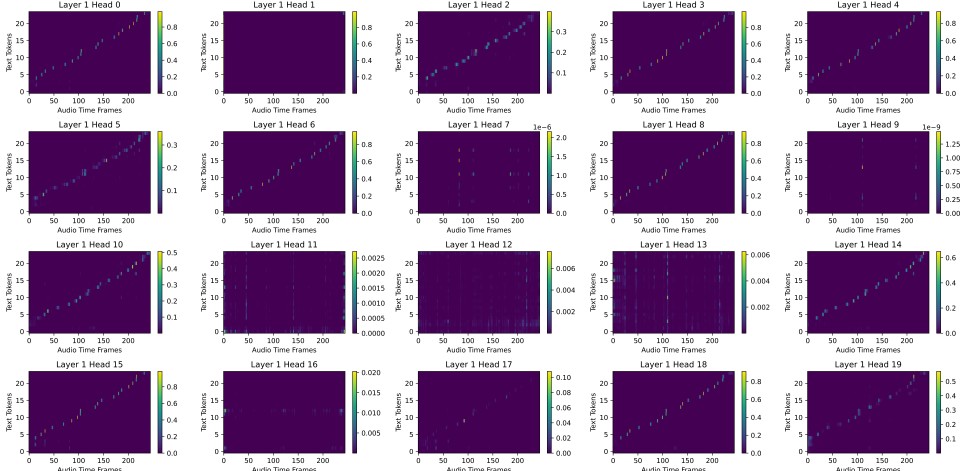

Figure 4: **The cross attention map of the last layer in our aggregator.** As illustrated, a lot of heads clearly demonstrate the text-speech aligned behavior.

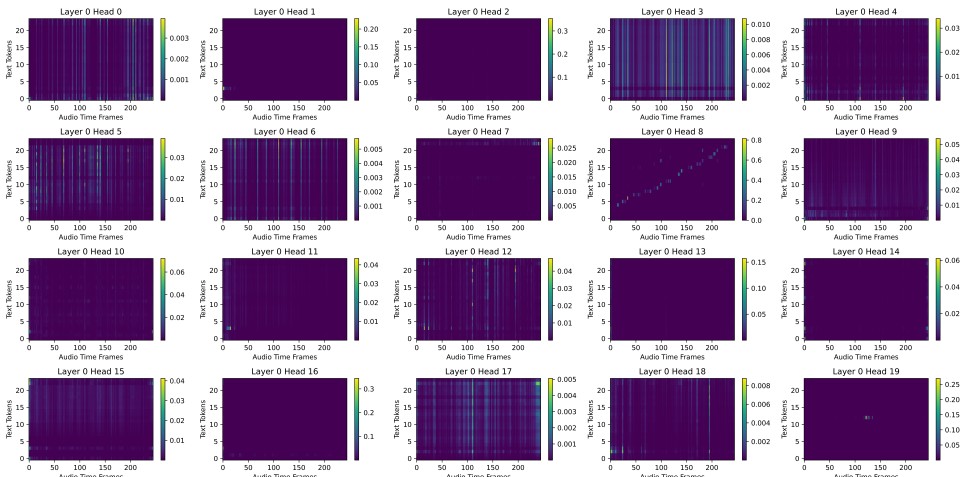

Figure 5: **The cross attention map of the first layer in our aggregator.** As illustrated, the behavior is quite different from the last layer in Figure 4. We observe some special heads: Head 8th showns clear alignment; while head 19th lights up at the silence part.

### A.1.3 DISCUSSION ON THE SELECTION OF SHALLOW HIDDEN LAYER

In Section 3.1.1, we propose using the shallow hidden representation from Whisper encoder as the key in our specialized cross attention, as it carries rich acoustic information. In practice, we select $l = 6$. In Table 4, this allows us to achieve a near-optimal reconstruction accuracy on the target S3 unit. To further justify this selection, we follow the analysis methodology of Pasad et al. (2021) and compute the Canonical Correlation Analysis (CCA; Hotelling (1992)) between each Whisper encoder layer and the target S3 unit embeddings. The resulting similarity curve is shown in Figure 6. The correlations peak at the 4-th to 8-th layers, indicating that these shallow layers encode representations most aligned with the S3 targets—supporting our design choice of using a shallow hidden state. Importantly, Table 4 also shows that while the choice of the shallow layer affects the attainable upper bound, it does not impose a strict limitation on our method: even using the final encoder layer, whose correlation to S3 is much lower, still yields reconstruction quality better than the text-only baseline.

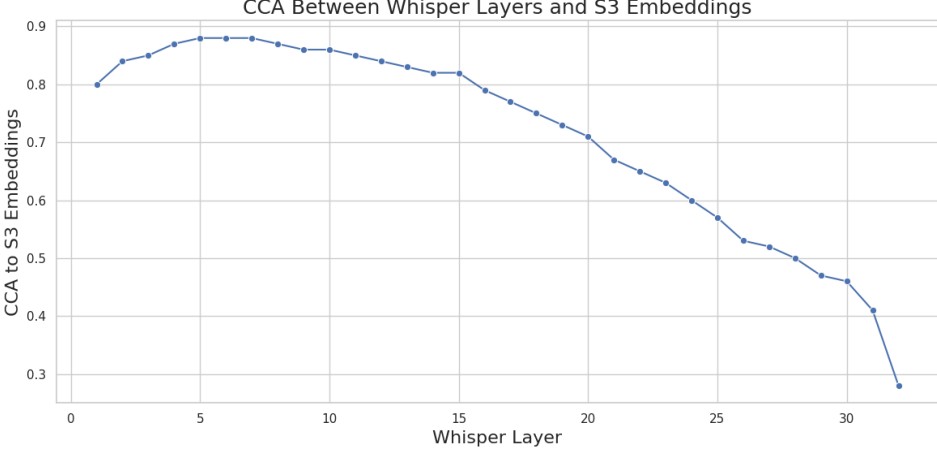

Figure 6: **Canonical Correlation Analysis (CCA) between each Whisper encoder layer and the S3 target embeddings.** Layers 4–8 exhibit the highest correlation with the S3 units, indicating that shallow representations contain the most target-relevant acoustic information.

Table 8: **The tokenizer robustness ablation study.** We apply four different level of noise, with signal-to-noise ratio (SNR) ranging from 20dB (almost clean) to 5dB (very noisy). Our method substantially achieves good reconstruction quality across all noise levels.

| Method | Bitrate | SNR=20dB (clean) | | | SNR=15dB | | | SNR=10dB | | | SNR=5dB (noisy) | | |
|---|---|---|---|---|---|---|---|---|---|---|---|---|---|
| | | WER ↓ | Sim. ↑ | Rank | WER ↓ | Sim. ↑ | Rank | WER ↓ | Sim. ↑ | Rank | WER ↓ | Sim. ↑ | Rank |
| Ground Truth | 256k | 2.3% | - | - | 2.5% | - | - | 3.6% | - | - | 9.2% | - | - |
| DAC[α] | 500 | 93.7% | 0.193 | 13 | 97.7% | 0.201 | 12 | 98.6% | 0.205 | 12 | 98.3% | 0.202 | 13 |
| | 1000 | 36.0% | 0.292 | 11 | 55.0% | 0.276 | 11 | 80.2% | 0.278 | 11 | 94.5% | 0.283 | 10 |
| | 12000 | 2.5% | 0.937 | 1 | 2.9% | 0.924 | 1 | 4.7% | 0.914 | 1 | 12.3% | 0.907 | 1 |
| DM-Codec[β] | 500 | 98.7% | 0.295 | 12 | 104.6% | 0.264 | 12 | 107.1% | 0.264 | 12 | 105.8% | 0.283 | 12 |
| | 1000 | 33.9% | 0.479 | 9 | 53.0% | 0.447 | 9 | 77.4% | 0.435 | 9 | 97.1% | 0.415 | 9 |
| | 4000 | 3.9% | 0.737 | 5 | 6.4% | 0.689 | 6 | 14.4% | 0.647 | 6 | 38.5% | 0.595 | 6 |
| SpeechTokenizer[γ] | 500 | 15.2% | 0.334 | 9 | 33.9% | 0.324 | 9 | 64.2% | 0.301 | 9 | 91.7% | 0.277 | 10 |
| | 2000 | 7.3% | 0.773 | 8 | 16.3% | 0.734 | 8 | 39.4% | 0.682 | 8 | 73.8% | 0.600 | 8 |
| | 4000 | 4.4% | 0.864 | 2 | 8.1% | 0.822 | 4 | 21.0% | 0.765 | 4 | 49.2% | 0.684 | 5 |
| BigCodec[δ] | 1040 | 10.1% | 0.829 | 7 | 17.8% | 0.785 | 7 | 33.5% | 0.718 | 7 | 63.0% | 0.625 | 6 |
| Mimi[ε] | 1000 | 5.1% | 0.804 | 6 | 7.8% | 0.772 | 5 | 14.7% | 0.726 | 4 | 33.6% | 0.673 | 4 |
| S3 token[ζ] | 600 | 3.9% | 0.860 | 2 | 5.2% | 0.841 | 2 | 8.1% | 0.815 | 2 | 16.7% | 0.779 | 3 |
| TASTE (ours) | ∼**150** | 4.8% | 0.842 | 4 | 5.3% | 0.830 | 3 | 6.9% | 0.815 | 2 | **11.1%** | 0.792 | **1** |

[α] Kumar et al. (2023); [β] Ahasan et al. (2025); [γ] Zhang et al. (2024); [δ] Xin et al. (2024b); [ε] Défossez et al. (2024); [ζ] Du et al. (2024a)

### A.1.4 ROBUSTNESS ABLATION ON TASTE TOKENIZER

To evaluate our tokenizer robustness, we have conducted controlled noise level experiments on speech reconstruction. Specifically, we add 4 different levels of white noise to the original waveform and then evaluate the reconstruction performance. The noise levels are defined by signal-to-noise ratio (SNR) ranging from 20dB (almost clean) to 5dB (very noisy). We report two metrics: ASR-WER as an indicator of reconstruction quality, and speaker similarity (Sim.) as a measure of reconstruction similarity. For ease of comparison, we also provide the overall rank of each tokenizer considering both metrics. The results in Table 8 show that our tokenizer remains stable across different noise levels, demonstrating strong robustness. Notably, TASTE achieves the best ASR-WER under the noisiest condition. This suggests that the underlying ASR system is not a limiting factor for the general applicability of TASTE; rather, it serves as an effective and reliable source of semantic information for the reconstruction.

### A.1.5 DETAILS ON SALMON AND STORYCLOZE

Our detailed results on SALMON and StoryCloze are reported in Table 9. The introductions of the two benchmarks—SALMON and StoryCloze—are described below.

**SALMON for Acoustic Evaluation** SALMON offers a comprehensive set of metrics designed to evaluate SLMs in multiple dimensions. In summary, each test sample consists of a *positive* sample and a *negative* sample. The *negative* sample differs from the *positive* sample by having some

Table 9: The evaluation results on SALMON and StoryCloze of different SLMs, and BG means background. We report likelihood-based accuracy on SALMON (acoustic aspect) and StoryCloze (semantic aspect). The baseline (S3 token) is conducted by joint speech-text modeling with the S3 token as speech tokenization.

| METHOD | LoRA | SALMON (ACOUSTIC CONSISTENCY) | | | | | | STORYCLOZE |
|---|---|---|---|---|---|---|---|---|
| | | Sentiment | Speaker | Gender | Room | BG (domain) | BG (rand.) | sSC / tSC |
| *Previous Work* | | | | | | | | |
| TWIST 1.3B ((Hassid et al., 2023)) | ✗ | 61.5±3.4 | 69.0±3.3 | 69.5±3.3 | 59.0±3.5 | 55.5±3.5 | 60.5±3.5 | 52.4±0.8 / 70.6±0.7 |
| TWIST 7B ((Hassid et al., 2023)) | ✗ | 61.5±3.4 | 71.0±3.2 | 70.0±3.2 | 62.0±3.4 | 55.5±3.5 | 60.5±3.5 | 55.3±0.8 / 74.1±0.7 |
| Spirit LM ((Nguyen et al., 2025)) | ✗ | 54.5±3.5 | 69.5±3.3 | 67.0±3.3 | 54.5±3.5 | 53.5±3.5 | 55.5±3.5 | 61.0±0.8 / 82.9±0.6 |
| Spirit LM Expr. ((Nguyen et al., 2025)) | ✗ | 73.5±3.1 | 81.0±2.8 | 85.0±2.5 | 54.5±3.5 | 56.0±3.5 | 64.0±3.4 | 56.9±0.8 / 75.4±0.7 |
| *Ours* | | | | | | | | |
| Baseline (S3 token) | ✓ | 49.5±3.5 | 48.8±3.5 | 48.8±3.5 | 49.5±3.5 | 55.3±3.5 | 49.5±3.5 | 54.4±0.8 / 63.0±0.8 |
| TASLM 1B (token) | ✓ | 59.0±3.5 | 68.0±3.3 | 70.5±3.2 | 61.0±3.4 | 52.0±3.5 | 54.0±3.5 | 64.2±0.8 / 88.9±0.5 |
| TASLM 1B (embedding) | ✓ | 57.5±3.5 | 67.0±3.3 | 75.5±3.0 | 50.0±3.5 | 47.0±3.5 | 49.0±3.5 | 64.0±0.8 / 89.5±0.5 |

segments altered. These alterations include changes in speaker, gender, environment (e.g., room acoustics), or sentiment in the middle of the utterance. The SLM serves as an anomaly detector that aims to distinguish between the pairs of *positive* and *negative* samples. The distinction is based on the likelihood score given by each SLM, which is then evaluated with the overall precision between the ground truth and the prediction.

**StoryCloze for Semantic Evaluation**   To evaluate the SLMs' ability to comprehend semantic coherence and logical reasoning, we employ the spoken version of StoryCloze test (sSC) and the Topic StoryCloze test (tSC) assembled by Hassid et al. (2023). Assessment of narrative understanding involves presenting a four-sentence story setup, followed by two possible endings. These tasks require the model to select the most appropriate conclusion, thereby testing its grasp of causal and temporal relationships within a narrative. Similarly to SALMON, we measure the accuracy of the distinctions based on the likelihood scores.

**Discussion on SALMON and StoryCloze**   On SALMON, we observe that our TASLM falls short for background-related attributes (Room, Background) where the samples are added with environmental sounds (echoes, instruments, background noises from FSD50K). Since TASTE tokenization focuses on natural speech and has not being trained on audio with environmental sound and noise, it may not be able to convey such information. However, on the speech related attributes, such as gender and speaker, our TASLM performs much better and is comparable to other SLMs. On StoryCloze, our TASLM successfully retains its semantic capability with effective joint modeling on TASTE, leading to the best results among all pretrained SLMs.

### A.1.6   REPORT OF STANDARD DEVIATIONS

We report the standard deviations of our tables in the main text to allow further investigation.

Table 10: **Results with standard deviations of Table 1**

| Method | Bitrate | QUALITY | | | SIMILARITY | | | |
|---|---|---|---|---|---|---|---|---|
| | | WER ↓ | UTMOS | DNSMOS | ViSQOL | Drtn. Con. | Spkr. Sim. | MUSHRA |
| Ground Truth | 256k | 2.1%±0.07 | 4.09±0.32 | 3.84±0.26 | - | - | - | 76.6±15.9 |
| Encodec (Défossez et al., 2023) | 1500 | 5.1%±0.11 | 1.58±0.34 | 3.26±0.24 | 3.46±0.28 | 0.94±0.003 | 0.63±0.10 | - |
| | 3000 | 2.6%±0.08 | 2.35±0.53 | 3.48±0.25 | 3.81±0.27 | 0.96±0.002 | 0.78±0.07 | 25.6±18.6 |
| SpeechTokenizer (Zhang et al., 2024) | 500 | 5.2%±0.11 | 1.27±0.05 | 2.99±0.17 | 2.80±0.24 | 0.94±0.003 | 0.35±0.09 | - |
| | 2000 | 3.0%±0.08 | 3.56±0.43 | 3.60±0.28 | 3.65±0.22 | 0.97±0.002 | 0.80±0.06 | 53.9±22.9 |
| | 4000 | 2.5%±0.08 | 3.90±0.36 | 3.76±0.28 | 4.03±0.17 | 0.98±0.002 | 0.92±0.04 | - |
| Mimi (Défossez et al., 2024) | 1000 | 3.1%±0.09 | 3.60±0.37 | 3.60±0.30 | 3.62±0.26 | 0.96±0.002 | 0.82±0.06 | 67.6±19.8 |
| S3 token (topline) (Du et al., 2024a) | 600 | 3.0%±0.09 | 4.18±0.27 | 3.90±0.24 | 3.30±0.26 | 0.96±0.002 | 0.82±0.09 | 70.2±17.0 |
| Text-only (baseline) | ∼50 | 5.9%±0.11 | 4.31±0.16 | 4.11±0.22 | 2.44±0.23 | 0.57±0.006 | 0.78±0.09 | 42.6±27.1 |
| TASTE (ours) | ∼150 | 4.4%±0.11 | 4.29±0.18 | 4.10±0.22 | 3.05±0.26 | 0.91±0.003 | 0.80±0.08 | 68.3±17.1 |

Table 11: **Results with standard deviations of Table 2.**

| Method | Finetuned / base parameters | CONTINUATION | | | LIKELIHOOD | | |
|---|---|---|---|---|---|---|---|
| | | GPT-4o | UTMOS | Human | SALMON | StoryCloze | Overall |
| *Cascade* | | | | | | | |
| Cascade (LLaMA3.2-1B[α]) | - | 3.15±1.27 | 4.25±0.22 | 4.00±1.28 | - | - | - |
| Cascade (LLaMA2-7B[β]) | - | 3.43±1.27 | 4.25±0.25 | 3.98±1.29 | - | - | - |
| *Spoken LMs* | | | | | | | |
| TWIST 1.3B (Hassid et al., 2023) | 1.3B / 1.3B[θ] | 1.48±0.70 | 3.25±0.48 | 1.95±1.01 | 62.5±1.4 | 61.5±0.5 | 62.0±0.7 |
| TWIST 7B (Hassid et al., 2023) | 7B / 7B[γ] | 1.44±0.70 | 3.27±0.52 | 2.04±0.91 | 63.4±1.4 | 64.7±0.5 | 64.1±0.7 |
| Spirit LM (Nguyen et al., 2025) | 7B / 7B[β] | 2.79±1.06 | 3.41±0.19 | 2.38±0.81 | 59.1±1.4 | 72.0±0.5 | 65.6±0.7 |
| Spirit LM Expr. (Nguyen et al., 2025) | 7B / 7B[β] | 1.90±1.03 | 3.40±0.30 | 2.41±0.96 | 69.0±1.3 | 66.2±0.5 | 67.6±0.7 |
| Baseline (S3 token) | 45M / 1.3B[α] | 1.37±0.87 | 4.04±0.27 | 2.84±1.11 | 50.2±1.4 | 58.7±0.6 | 54.5±0.8 |
| TASLM 1B (token) | 45M / 1.3B[α] | 3.08±1.37 | 4.07±0.28 | 3.93±1.30 | 60.8±1.4 | 76.5±0.5 | 68.7±0.7 |
| TASLM 1B (embed.) | 45M / 1.3B[α] | 3.16±1.33 | 4.22±0.21 | 4.16±1.20 | 57.7±1.4 | 76.7±0.5 | 67.2±0.7 |

Base models: [α]LLaMA3.2-1B, [β]LLaMA2-7B, [γ]LLaMA-7B, [θ]OPT-1.3B

Table 12: **Results with standard deviations of Table 3.**

| Method | Mode | Web Q. | LLaMA-Q. |
|---|---|---|---|
| Mini-Omni 0.5B(T→T) | T | 21.3±0.9 | 39.0±2.8 |
| Mini-Omni 0.5B (Xie & Wu, 2024a) | T+A | 4.5±0.5 | 11.6±1.8 |
| Helium 7B (text) | T | 32.3±1.0 | 75.0±2.5 |
| Moshi 7B (Défossez et al., 2024) | T+A | 26.6±1.0 | 62.3±2.8 |
| LLaMA3.1-8B-Instruct | T | 60.4±1.1 | 71.7±2.6 |
| Llama-Omni-8B (Fang et al., 2024) | T+A | 35.5±1.1 | 67.3±2.7 |
| LLaMA3.2-1B[†] | T | 24.0±0.9 | 51.0±2.9 |
| TASLM 1B (embed.)[†] | T+A | 27.1±1.0 | 57.6±2.9 |

[†]We apply few-shot learning to facilitate question answering.

## A.2 TRAINING DETAILS

### A.2.1 HYPERPARAMETERS AND CONFIGURATIONS

We separate the training process into the two phases: *deriving TASTE tokenization* and *conducting spoken language modeling with TASTE*. In the tokenization phase, only the Aggregator, Quantizer, and the UnitDecoder is trainable. We use the Adam (Kingma, 2015) optimizer and the learning rate is set to 0.0016. The batch size is set to 160 seconds on each of the 8 NVIDIA A6000 GPUs we used. Note that in the first 2 epochs the quantization is not applied. From the beginning of the third epoch, quantization is applied and the Quantizer starts to be updated. We train the TASTE tokenizer for 5 epochs, which takes about 2 days for learning, with the learning rate gradually decayed.

As for the spoken language modeling training phase, we use the AdamW (Loshchilov & Hutter, 2019) optimizer, the Consine scheduler with the learning rate set to 1e-5. We use 8 Nvidia A6000 GPUs for training. The total batch size summation over the GPUs is set to 768 samples with the gradient accumulation steps set to 2. To reduce the memory overhead and the computational cost, we employ `bfloat16` mixed precision during training. Tools such as DeepSpeed (Rasley et al., 2020) and Liger Kernel (Hsu et al., 2024) are also applied to speed up the fine-tuning process of the SLM.

## A.3 EVALUATION DETAILS

### A.3.1 HUMAN EVALUATION

We conduct human listening tests through Amazon Mechanical Turk. In each experiment, we randomly select the same 20 samples from each method; and for each sample we collect more than 10 evaluation scores across different human evaluators.

**MUSHRA** In Table 1, we have shown our result of the MUSRHA human listening test (Series, 2014). Following Zhang et al. (2024), we conduct the evaluation with a hidden reference but without a lowerpass-filtered anchor. We instruct evaluators to rate the perceptual quality of the given samples with respect to the ground truth on a scale of 1 to 100.

**Speech Continuation MOS** In Table 2, we mention that we have conducted the human listening test to evaluate the overall performance of the speech continuations. Here, we present the instruction for human speech continuation MOS evaluation as follows:

---

**Instruction for Human Speech Continuation MOS Evaluation**

In this test, each sample will contain a short audio clip called "prompt" (3 seconds) and a longer audio clip called "prompt+continuation" (∼15 seconds).
You will be asked to rate the speech quality of the "prompt+continuation" audio clip, specifically focus on the "continuation" part.
The rating should be based on how likely you think that the long audio is a proper continuation of the "prompt" audio.
Specifically, the rating should be based on the following scale:

1: Bad - The "continuation" is not distinguishable or not natural.
2: Poor - The "continuation" is 25% distinguishable.
3: Fair - The "continuation" is 50% distinguishable and natural.
4: Good - The "continuation" is 75% distinguishable and natural.
5: Excellent - The "continuation" is distinguishable, meaningful, and natural.

**Distinguishable** means that the words in the "continuation" is distinguishable.
**Natural** means that the "continuation" sounds like a real human voice and a natural continuation of the prompt without considering the content of the speech.
**Meaningful** means that you can not only distinguish the words but also understand the meaning of the whole "prompt+continuation".

---

### A.3.2 GPT-4O FOR MOS EVALUATION

As introduced in Section 5.1.2, we use GPT-4o to assign MOS scores to the speech continuation results (Chiang & Lee, 2023; Lin et al., 2024). Here, we describe the detailed procedure. First, `whisper-large-v3` is applied to transcribe the generated speech. Then, given the transcription, the text content from the prompt audio, and the instruction template, GPT-4o can produce a score between 1 and 5. The instruction template is provided below:

---

**Instruction Prompt for GPT-4o MOS Evaluation**

```
The task is evaluating the relevance and likelihood of the
predicted text continuation, given the text prompt.  You should
also consider whether the meaning of the text continuation is
making sense.  The text prompt is:

"{prompt}"
, and the text continuation is :
"{content}"

You must give an overall rating from 1 to 5.  The rating guideline
is as below:

1:  The text continuation is very unlikely and irrelevant to the
text prompt.
2:  The text continuation is unlikely and marginally relevant to
the text prompt.
3:  The text continuation is moderately likely and relevant to the
text prompt.
4:  The text continuation is likely and relevant to the text
prompt.
5:  The text continuation is very likely and highly relevant.

You should take the following steps to provide the score:
First:  briefly analyze the sample with the above definition.
Second:  MUST follow the output format as:  I would rate the score
as _
```

---

### A.4 TACKLING THE VOCABULARY MISMATCH

The vocabulary mismatch problem lies in the fact that the vocabulary sets are different between the ASR and the LLM, and TASTE is aligned with the text transcription tokens from ASR. Consider that given a text transcription $v$ and the vocabulary sets of ASR and LLM denoted as $\mathbb{V}^{\text{asr}}$ and $\mathbb{V}^{\text{llm}}$, the ASR tokenized sequence $v^{\text{asr}} = [v_1^{\text{asr}}, v_2^{\text{asr}}, \ldots, v_N^{\text{asr}}], v_i^{\text{asr}} \in \mathbb{V}^{\text{asr}}$ and the LLM tokenized sequence $v^{\text{llm}} = [v_1^{\text{llm}}, v_2^{\text{llm}}, \ldots, v_M^{\text{llm}}], v_i^{\text{llm}} \in \mathbb{V}^{\text{llm}}$ can be different in terms of token ids and sequence lengths. Since the TASTE token and embedding are aligned with $v^{\text{asr}}$, we need to derive a method to align them with $v^{\text{llm}}$ for text-aligned speech-text modeling. Notice that $v^{\text{asr}}$ and $v^{\text{llm}}$ both represent $v$, we propose to mitigate the issue through word-level *grouping*, *averaging*, and *aligning*, detailed in Algorithm 1. By crafting TASTE speech tokenization into the word level, we are able to align it with the text tokens of the LLM, denoted as $\tilde{q}, \tilde{z}$. In practice, we also adopt the word-level averaging technique during the TASTE tokenization training phase, ensuring that the word-level TASTE tokenization facilitates high-quality reconstruction.

---

**Algorithm 1** Aligning TASTE with LLM Tokenization via Word-Level Techniques

---

1: **Initialization:**

Text transcription $\boldsymbol{v} = [\text{word}_1, \text{word}_2, \ldots, \text{word}_W]$

ASR tokens of the transcription $\boldsymbol{v}^{\text{asr}} = [v_1^{\text{asr}}, v_2^{\text{asr}}, \ldots, v_N^{\text{asr}}]$

TASTE embedding $\hat{\boldsymbol{z}} = [\hat{z}_1, \hat{z}_2, \ldots, \hat{z}_N]$

LLM tokens of the transcription $\boldsymbol{v}^{\text{llm}} = [v_1^{\text{llm}}, v_2^{\text{llm}}, \ldots, v_M^{\text{llm}}]$

2: **procedure** WORDLEVELGROUPING($\boldsymbol{v}, \boldsymbol{v}^{\text{asr}}, \hat{\boldsymbol{z}}, \boldsymbol{v}^{\text{llm}}$)

3:      Since $\boldsymbol{v}^{\text{asr}}$ is a token sequence represents $\boldsymbol{v}$, we can easily group it by words:

4:      $\boldsymbol{v}_{\text{grouped}}^{\text{asr}} \leftarrow [\underbrace{(v_1^{\text{asr}}, v_2^{\text{asr}}, v_3^{\text{asr}})_1}_{\text{word}_1}, \underbrace{(v_4^{\text{asr}})_2}_{\text{word}_2}, \ldots, \underbrace{(v_{N-1}^{\text{asr}}, \boldsymbol{v}_N^{\text{asr}})_W}_{\text{word}_W}]$     $\triangleright$ Group $\boldsymbol{v}^{\text{asr}}$ by the words of $\boldsymbol{v}$

5:      With the word-level grouping from $\boldsymbol{v}_{\text{grouped}}^{\text{asr}}$, we can group TASTE embedding $\hat{\boldsymbol{z}}$ as well:

6:      $\hat{\boldsymbol{z}}_{\text{grouped}} \leftarrow [(\hat{z}_1, \hat{z}_2, \hat{z}_3)_1, (\hat{z}_4)_2, \ldots, (\hat{z}_{N-1}, \hat{z}_N)_W]$

7:      Finally, we can group $\boldsymbol{v}^{\text{llm}}$ following the similar procedure of grouping $\boldsymbol{v}^{\text{asr}}$:

8:      $\boldsymbol{v}_{\text{grouped}}^{\text{llm}} \leftarrow [\underbrace{(v_1^{\text{llm}}, v_2^{\text{llm}})_1}_{\text{word}_1}, \underbrace{(v_3^{\text{llm}}, v_4^{\text{llm}})_2}_{\text{word}_2}, \ldots, \underbrace{(v_{M-2}^{\text{llm}}, v_{M-1}^{\text{llm}}, v_M^{\text{llm}})_W}_{\text{word}_W}]$

9:      Due to the *vocabulary mismatch*, the grouping of $\boldsymbol{v}_{\text{grouped}}^{\text{llm}}$ is different from $\boldsymbol{v}_{\text{grouped}}^{\text{asr}}, \hat{\boldsymbol{z}}_{\text{grouped}}$.

10: **end procedure**

11: **procedure** WORDLEVELAVERAGING($\hat{\boldsymbol{z}}_{\text{grouped}}$)

12:      $\bar{\boldsymbol{z}} \leftarrow []$                                        $\triangleright$ Initialize a new sequence

13:      **for** word group index $i \leftarrow 1$ to $W$ **do**

14:          word group $(\hat{z}_j, \ldots, \hat{z}_k) \leftarrow \hat{\boldsymbol{z}}_{\text{grouped}}[i]$

15:          $\bar{z}_{[j:k]} \leftarrow \text{Average}((\hat{z}_j, \ldots, \hat{z}_k))$                $\triangleright$ Average the word group

16:          **append** $\bar{z}_{[j:k]}$ **to** $\bar{\boldsymbol{z}}$

17:      **end for**

18:      Resulting in word-level TASTE embedding $\bar{\boldsymbol{z}} \in \mathbb{R}^{W \times d_z}$, where $W$ is the word length of $v$.

19: **end procedure**

20: **procedure** ALIGNWORDLEVELEMBEDDINGWITHLLM($\bar{\boldsymbol{z}}, \boldsymbol{v}_{\text{grouped}}^{\text{llm}}$)

21:      $\tilde{\boldsymbol{z}} \leftarrow []$                                        $\triangleright$ Initialize a new sequence

22:      **for** word group index $i \leftarrow 1$ to $W$ **do**

23:          word group $(v_j^{\text{llm}}, \ldots, v_k^{\text{llm}}) \leftarrow \boldsymbol{v}_{\text{grouped}}^{\text{llm}}[i]$

24:          $M \leftarrow \text{Length}((v_j^{\text{llm}}, \ldots, v_k^{\text{llm}}))$        $\triangleright$ Get the length of the word group.

25:          **for** $m \leftarrow 1$ to $M$ **do**           $\triangleright$ add $M \times \bar{z}[i]$ into the aligned sequence $\tilde{\boldsymbol{z}}$

26:              **append** $\bar{z}[i]$ **to** $\tilde{\boldsymbol{z}}$

27:          **end for**

28:      **end for**

29: **end procedure**

30: **return** The LLM-aligned word-level TASTE embedding $\tilde{\boldsymbol{z}}$ and its codes form $\tilde{\boldsymbol{q}}$

---

## A.5 THE WEIGHTED SUM MECHANISM FOR MODALITY FUSION

The speech decoder takes the text-aligned speech embedding $\hat{\boldsymbol{z}}$ and the text embedding $\boldsymbol{v}$ as input conditions. As depicted in Figure 2, in practice we perform a *weighted sum* mechanism to obtain the final fused embedding $\boldsymbol{z}_{\text{joint}}$. The weights to fuse the two modalities are represented by two learnable parameters, denoted as $w_{\text{sp}}$ and $w_{\text{txt}}$. The whole process of modality fusion can be denoted as follows:

$$\hat{\boldsymbol{z}}', \boldsymbol{v}' = \text{normalize}(\hat{\boldsymbol{z}}), \text{normalize}(\boldsymbol{v}), [p_{\text{sp}}, p_{\text{txt}}] = \text{softmax}([w_{\text{sp}}, w_{\text{txt}}]), z_{\text{joint}} = p_{\text{sp}} \cdot \hat{\boldsymbol{z}}' + p_{\text{txt}} \cdot \boldsymbol{v}'.$$

## A.6 DISCUSSION ON THE USAGE OF LLM

We discuss our usage of LLM following the conference's policy. We use an AI assistant (ChatGPT specifically) to polish English prose, including grammar correction, wording refinements, consistent terminology and hyphenation, and minor restructuring for clarity and flow. The assistant suggests alternative phrasings, section bridges, and standard disclosure/impact wording based on author-provided content. It does not generate novel ideas, claims, analyses, figures, code, or results, and it does not access proprietary data. All technical content and conclusions are our own, and we review and edit all AI-assisted text and take full responsibility for the final manuscript.

