# OpenReview forum: "TASTE: Text-Aligned Speech Tokenization and Embedding for Spoken Language Modeling"
_ICLR.cc/2026/Conference — ICLR 2026 Poster_

### Official Review · Reviewer_FzoP · 2025-10-30

**Soundness:** 3
**Presentation:** 3
**Contribution:** 3
**Rating:** 6
**Confidence:** 4

**Summary:**

This work proposes TASTE, a speech tokenizer that enables the alignment with semantic text tokens. It can generate highly compressed audio tokens via minimization of the speech reconstruction loss. The experiments are performed to show that TASTE successfully reconstructs the audio and helps LLMs easily finetuned to be SLMs (Spoken Language Models).

**Strengths:**

1. This work proposes a novel highly compressible speech tokenizer named TASTE that can reduce the modality gap between text and audio pairs. This is practically useful to reduce the length of the token sequence for a given audio with little loss of reconstruction quality.

2. TASTE helps finetuning LLMs to SLMs via Low-Rank Adaptation (LoRA). This is practically useful to speed up the development of SLMs or omni-modal LLMs.

**Weaknesses:**

1. One of the most practical uses of audio tokenizers may be training of SLMs, as this work experimented.
- Thus, it may need to be studied whether the proposed tokenizers can lead to SOTA performance of SLMs.
- For example, TASLM’s performance can be compared with that of SOTA SLM models such as LLaMA-Omni 2.

2. The generalization of this tokenizer should be discussed.
- What if the domains of text and audio corpora are changed? Do we need to train again?
- Since the proposed tokenizers are forced to learn the joint representation of both “semantics” and “phonetics”, the learned tokenizer may be much more variable according to the training sets than the one learned only focusing on “phonetics”.

**Questions:**

Please refer to the Weaknesses.

---

> ### Author Response · Authors · 2025-11-21
>
> We thank the reviewer for the thorough and constructive comments. Please find the response to your questions below.
>
> *[W1] One of the most practical uses of audio tokenizers may be training of SLMs, as this work experimented. Thus, it may need to be studied whether the proposed tokenizers can lead to SOTA performance of SLMs. For example, TASLM’s performance can be compared with that of SOTA SLM models such as LLaMA-Omni 2.*
>
> > Thank you for raising this point. We have demonstrated that our tokenizer can lead to effective pretrained SLM. The evaluation follows typical pretrained SLM evaluation and the results demonstrate the effectiveness. Extending it to more user-friendly systems (e.g., dialogue system, instruction following system like LLaMA-Omni) remains promising future work that requires further **post-training**, which may not be covered by this work due to the space and computational limitation.
>
> *[W2] The generalization of this tokenizer should be discussed.*
>
> > We are currently conducting a controlled experiment to further investigate the robustness of our tokenization. We would like to update the results once we have finished.
>
> *What if the domains of text and audio corpora are changed? Do we need to train again?*
>
> > We argue that our tokenization is trained on clean (LibriTTS) and in-the-wild (Emilia) combined corpora, which covers a wide variety of speech domains. To this end, we believe that the model is of sufficient robustness. Even if the dataset changes, one could finetune from our initial released checkpoint to reduce the computing overhead.

---

> ### Author Response · Authors · 2025-12-02
>
> Follow up on: *[W2] The generalization of this tokenizer should be discussed.*
> > We have finished the robustness ablation study of our tokenizer to address the reviewer's concern (*[W2]* in the previous official comment). The results are presented below.
>
> > **Table R5-1**. The robusness ablatioin study of our tokenization. We introduce different levels of white noise to evaluate how our tokenizer performs under varying noise conditions. Here, SNR denotes the signal-to-noise ratio, ranging from 20 dB (nearly clean) to 5 dB (very noisy). We report two metrics: ASR-WER as an indicator of reconstruction quality, and speaker similarity (SIM) as a measure of reconstruction similarity. For ease of comparison, we also provide the overall rank of each tokenizer considering both metrics.
>
> >|                     |           | ｜  |  SNR=20dB   | (almost |  clean)  | ｜  |  SNR=15dB   | (slightly |  noisy)  | ｜  |  SNR=10dB   | (fairly |  noisy)  | ｜  |   SNR=5dB   |  (very  |  noisy)  |
> >|:------------------- | ---------:|:---:|:-----------:|:-------:|:--------:|:---:|:-----------:|:---------:|:--------:|:---:|:-----------:|:-------:|:--------:|:---:|:-----------:|:-------:|:--------:|
> >| **Method**          |   **bps** | ｜  | **ASR-WER** | **SIM** | **Rank** | ｜  | **ASR-WER** |  **SIM**  | **Rank** | ｜  | **ASR-WER** | **SIM** | **Rank** | ｜  | **ASR-WER** | **SIM** | **Rank** |
> >| **groud-truth**     |      256k | ｜  |    2.3%     |   \-    |          | ｜  |    2.5%     |    \-     |          | ｜  |    3.6%     |   \-    |          | ｜  |    9.2%     |   \-    |          |
> >| **DAC**             |       500 | ｜  |    93.7%    |  0.193  |    13    | ｜  |    97.7%    |   0.201   |    12    | ｜  |    98.6%    |  0.205  |    12    | ｜  |    98.3%    |  0.202  |    13    |
> >| **DAC**             |      1000 | ｜  |    36.0%    |  0.292  |    11    | ｜  |    55.0%    |   0.276   |    11    | ｜  |    80.2%    |  0.278  |    11    | ｜  |    94.5%    |  0.283  |    10    |
> >| **DAC**             |     12000 | ｜  |    2.5%     |  0.937  |    1     | ｜  |    2.9%     |   0.924   |    1     | ｜  |    4.7%     |  0.914  |    1     | ｜  |    12.3%    |  0.907  |    1     |
> >| **DM-Codec**        |       500 | ｜  |    98.7%    |  0.295  |    12    | ｜  |   104.6%    |   0.264   |    12    | ｜  |   107.1%    |  0.264  |    12    | ｜  |   105.8%    |  0.283  |    12    |
> >| **DM-Codec**        |      1000 | ｜  |    33.9%    |  0.479  |    9     | ｜  |    53.0%    |   0.447   |    9     | ｜  |    77.4%    |  0.435  |    9     | ｜  |    97.1%    |  0.415  |    9     |
> >| **DM-Codec**        |      4000 | ｜  |  3.9%   |  0.737  | 5  | ｜  |    6.4%     |   0.689   |    6 | ｜  |    14.4%    |  0.647  |    6 | ｜  |    38.5%    |  0.595  |    6     |
> >| **speechtokenizer** |       500 | ｜  | 15.2%|  0.334  |    9     | ｜  |  33.9% |   0.324   |    9     | ｜  |    64.2%    |  0.301  |    9     | ｜  |    91.7%    |  0.277  |    10    |
> >| **speechtokenizer** |      2000 | ｜  |    7.3%     |  0.773  |    8     | ｜  |  16.3%  |   0.734   |    8     | ｜  |    39.4%    |  0.682  |    8     | ｜  |    73.8%    |  0.600  |    8     |
> >| **speechtokenizer** |      4000 | ｜  |    4.4%     |  0.864  |    2     | ｜  |    8.1%     |   0.822   |    4     | ｜  |    21.0% |  0.765  |    4     | ｜  |    49.2%    |  0.684  |    5     |
> >| **BigCodec**        |      1040 | ｜  |    10.1%    |  0.829  |    7     | ｜  |    17.8%    |   0.785   |    7     | ｜  |    33.5% |  0.718  |    7     | ｜  |    63.0%    |  0.625  |    6     |
> >| **mimi**            |      1000 | ｜  |    5.1%     |  0.804  |    6     | ｜  |    7.8%     |   0.772   |    5     | ｜  |    14.7%    |  0.726  |    4     | ｜  |    33.6%    |  0.673  |    4     |
> >| **s3 token**        |       600 | ｜  |    3.9%     |  0.860  |    2     | ｜  |    5.2%     |   0.841   |    2     | ｜  |    8.1%     |  0.815  |    2     | ｜  |    16.7%    |  0.779  |    3     |
> >| **TASTE**           | **\~153** | ｜  |    4.8%     |  0.842  |  **4**   | ｜  |    5.3%     |   0.830   |  **3**   | ｜  |    6.9%     |  0.815  |  **2**   | ｜  |  **11.1%**  |  0.792  |  **1**   |
>
> > The results show that **our tokenizer remains stable across different noise levels, demonstrating strong robustness**.

---

### Official Review · Reviewer_ajFq · 2025-10-31

**Soundness:** 3
**Presentation:** 3
**Contribution:** 3
**Rating:** 6
**Confidence:** 4

**Summary:**

The paper introduces TASTE, a text-aligned speech tokenization and embedding method aimed at simplifying joint text–speech modeling. Given a speech–text pair, a frozen Whisper encoder provides shallow and last-layer features, an attention-based aggregator uses text tokens as queries over these features to produce a text-length speech representation, which is discretized with RVQ, and finally, a decoder reconstructs speech from the text and the aligned speech embedding for a reconstruction objective.

**Strengths:**

- The key contribution is to align speech tokens to the text sequence during tokenization and show that this enables simpler, stronger joint modeling at low bitrate. The idea is clean and the empirical gains on continuation with a small LM are practically meaningful for SLM research.

-  TASLM (1.3B + LoRA) outperforms prior 7B SLMs on continuation (GPT-4o, UTMOS, Human) and is competitive on SALMON/StoryCloze. The LoRA-only fine-tuning and word-level handling of ASR/LLM vocab mismatch increase practical relevance.

- ASR robustness ablations (GT vs ASR transcripts; Whisper vs Parakeet) show limited sensitivity, and the word-swap editing demo nicely evidences alignment of paralinguistics to lexical units.

**Weaknesses:**

- While key idea of the proposed method is clean, novelty is rather moderate. Text-speech cross-attention has been used for fine-grained TTS control, and recent joint SLMs alleviate length mismatch via interleaving (e.g., Spirit-LM) or lower-frequency tokenizers (e.g., moshi).

- Key "semantic MOS" relies on GPT-4o over ASR transcripts. Despite ablations, this double-black-box can amplify mistakes, such as prosody -> ASR error -> GPT score. Human semantic judgments or text-only LM plausibility on ASR outputs would be a stronger counterpoint, and code-switch/noisy-ASR conditions would probe robustness more realistically.

- The autogressive "weighted sum" between text and aligned speech in the unit decoder is under-specified. I would suggest the authors include equations/ablation to clarify.

**Questions:**

- Please detail the decoder fusion: where are the gates/weights applied between $v$ and $\hat{z}$, and how are they trained and stabilized?

- Can you report out-of-domain continuation and SALMON/StoryCloze on non-LibriVox corpora, and include dedup checks between LibriTTS and LibriSpeech?

- What are latency/memory costs for tokenization/decoding, and can TASTE stream in real time?

**Details Of Ethics Concerns:**

The authors have not addressed ethical statement. The proposed method enables convincing voice continuation/editing that could aid impersonation.

---

> ### Author Response · Authors · 2025-11-21
>
> We thank the reviewer for the thorough and constructive comments. Please find the response to your questions below.
>
> *[W1] While key idea of the proposed method is clean, novelty is rather moderate. Text-speech cross-attention has been used for fine-grained TTS control, and recent joint SLMs alleviate length mismatch via interleaving (e.g., Spirit-LM) or lower-frequency tokenizers (e.g., moshi).*
>
> > As discussed in Related Work, we do not consider our main novelty as using text-speech cross-attention. Instead, we attribute it to the concept of deriving joint tokenization for effective joint SLM. In our work, we completely alleviate the length mismatch problem in the early tokenization stage; so that we would use the smallest momentum to effectively model it. The idea behind is never proposed or explored before, and we wish our work encourages further research on more effective joint tokenization for generative modeling.
>
> *[W2] Key "semantic MOS" relies on GPT-4o over ASR transcripts. Despite ablations, this double-black-box can amplify mistakes, such as prosody -> ASR error -> GPT score. Human semantic judgments or text-only LM plausibility on ASR outputs would be a stronger counterpoint, and code-switch/noisy-ASR conditions would probe robustness more realistically.*
>
> > We understand your concerns. However, since our human evaluation combinedly evaluates the overall quality of speech, the human semantic judgment is covered by such evaluation as well. Please refer to the Appendix A.3.1 for the human speech MOS evaluation. On the other hand, we are not quite sure about the meaning of text-only LM plausibility on ASR outputs. In our GPT-4o evaluation, we have already treated GPT-4o as a text-only LLM that takes ASR outputs for semantic evaluation. The prompt template can be found in Appendix A.3.2. Please feel free to clarify if we have any misunderstanding. As for the robustness conditions, we are currently conducting experiments regarding it. We will update the results here once we have finished.
>
> *[W3] The autogressive "weighted sum" between text and aligned speech in the unit decoder is under-specified. I would suggest the authors include equations/ablation to clarify.*
>
> Thank you for pointing this out. We provide a more precise description of the procedure below.
> Given a text embedding sequence $v$ and the corresponding TASTE speech embedding $z$, we first apply layer normalization to obtain the normalized representations $v'$ and $z'$. We then introduce two trainable scalar weights, $a$ and $b$, which are normalized through a softmax to ensure that the contributions of the text and speech components form a convex combination. The joint embedding is obtained via a weighted sum:
> $v', z' = \mathrm{normalize}(v),\ \mathrm{normalize}(z)$,
> $[a', b'] = \mathrm{softmax}([a, b])$,
> $z_{\mathrm{joint}} = a' \cdot v' + b' \cdot z'$.
> This produces a sequence of fused representations that combines both textual and paralinguistic information in a learnable and stable manner. In practice, the learned weights (a, b) are initialized by (1, 1). After training, they become (0.68, 1.32).
> We have included this detail into our revision in Appendix A.2.2.
>
> *[Q1] Please detail the decoder fusion: where are the gates/weights applied between, and how are they trained and stabilized?*
>
> > Please see [W3]
>
> *[Q2] Can you report out-of-domain continuation and SALMON/StoryCloze on non-LibriVox corpora, and include dedup checks between LibriTTS and LibriSpeech?*
>
> > We appreciate the suggestion. However, rerunning the whole speech continuation and the evaluation on a different corpora requires time and computing resources, we regret that we might not be able to complete this experiment during the rebuttal period. We will try our best to provide the results as soon as possible. As for SALMON/StoryCloze, they are benchmarks that have their own dataset.
>
> > On the other hand, we have empirically verified that there is no overlapping file between the training set of LibriTTS and the testing set of LibriSpeech we use. The paper of LibriTTS also says that “It has the same speakers and subset split as the LibriSpeech corpus”. Thus, this should not cause any issue.

---

> ### Author Response · Authors · 2025-11-21
>
> *[Q3] What are latency/memory costs for tokenization/decoding, and can TASTE stream in real time?*
>
> >Thank you for raising this. We evaluate the latency of TASLM on an A6000 GPU. The TASLM pipeline consists of three major components: (1) TASTE tokenization on input audio, (2) spoken language model (SLM) generation, and (3) TASTE de-tokenization. The latency of each stage is approximately:
> >(1) TASTE Tokenization: (0.089 + 0.78 × input_length) seconds
> >(2) SLM Generation:: 0.0138 × output_length seconds
> >(3) TASTE De-tokenization: 0.33 × output_length seconds
> Assuming a streaming setup where 1 second of audio is processed and 1 second of output is generated per step, the overall end-to-end latency is approximately 1.22 seconds. This latency is comparable to commercial systems such as ChatGPT Voice Mode and Grok but excluding network delays.
> Currently, the primary bottleneck for processing speed lies in the TASTE tokenization and de-tokenization components. While this work focuses on demonstrating the conceptual validity of the approach, both of these components can be significantly accelerated through streaming architectures in future engineering implementations.
>
> >As for memory consumption, we argue that TASTE is in fact advantageous. The memory cost of both tokenization and detokenization is fixed and negligible compared to the downstream SLM. The dominant memory bottleneck in spoken language modeling arises from the KV cache, whose size grows linearly with sequence length during autoregressive decoding. By aligning speech tokens to textual granularity, TASTE significantly reduces the sequence length of speech inputs. This leads to a large reduction in KV-cache memory and allows the SLM to process much longer contexts under the same memory budget.
>
> *[E] The authors have not addressed ethical statement. The proposed method enables convincing voice continuation/editing that could aid impersonation.*
>
> > Thank you for pointing this out. We have revised the manuscript to include the ethical statement in our revision.

---

> ### Author Response · Authors · 2025-12-02
>
> Follow up on: *[W2] ...noisy-ASR conditions would probe robustness more realistically*
> > We have finished the robustness ablation study of our tokenizer to address the reviewer's concern (*[W2]* in the previous official comment). The results are presented below.
>
> > **Table R4-1**. The robusness ablatioin study of our tokenization. We introduce different levels of white noise to evaluate how our tokenizer performs under varying noise conditions. Here, SNR denotes the signal-to-noise ratio, ranging from 20 dB (nearly clean) to 5 dB (very noisy). We report two metrics: ASR-WER as an indicator of reconstruction quality, and speaker similarity (SIM) as a measure of reconstruction similarity. For ease of comparison, we also provide the overall rank of each tokenizer considering both metrics.
>
> >|                     |           | ｜  |  SNR=20dB   | (almost |  clean)  | ｜  |  SNR=15dB   | (slightly |  noisy)  | ｜  |  SNR=10dB   | (fairly |  noisy)  | ｜  |   SNR=5dB   |  (very  |  noisy)  |
> >|:------------------- | ---------:|:---:|:-----------:|:-------:|:--------:|:---:|:-----------:|:---------:|:--------:|:---:|:-----------:|:-------:|:--------:|:---:|:-----------:|:-------:|:--------:|
> >| **Method**          |   **bps** | ｜  | **ASR-WER** | **SIM** | **Rank** | ｜  | **ASR-WER** |  **SIM**  | **Rank** | ｜  | **ASR-WER** | **SIM** | **Rank** | ｜  | **ASR-WER** | **SIM** | **Rank** |
> >| **groud-truth**     |      256k | ｜  |    2.3%     |   \-    |          | ｜  |    2.5%     |    \-     |          | ｜  |    3.6%     |   \-    |          | ｜  |    9.2%     |   \-    |          |
> >| **DAC**             |       500 | ｜  |    93.7%    |  0.193  |    13    | ｜  |    97.7%    |   0.201   |    12    | ｜  |    98.6%    |  0.205  |    12    | ｜  |    98.3%    |  0.202  |    13    |
> >| **DAC**             |      1000 | ｜  |    36.0%    |  0.292  |    11    | ｜  |    55.0%    |   0.276   |    11    | ｜  |    80.2%    |  0.278  |    11    | ｜  |    94.5%    |  0.283  |    10    |
> >| **DAC**             |     12000 | ｜  |    2.5%     |  0.937  |    1     | ｜  |    2.9%     |   0.924   |    1     | ｜  |    4.7%     |  0.914  |    1     | ｜  |    12.3%    |  0.907  |    1     |
> >| **DM-Codec**        |       500 | ｜  |    98.7%    |  0.295  |    12    | ｜  |   104.6%    |   0.264   |    12    | ｜  |   107.1%    |  0.264  |    12    | ｜  |   105.8%    |  0.283  |    12    |
> >| **DM-Codec**        |      1000 | ｜  |    33.9%    |  0.479  |    9     | ｜  |    53.0%    |   0.447   |    9     | ｜  |    77.4%    |  0.435  |    9     | ｜  |    97.1%    |  0.415  |    9     |
> >| **DM-Codec**        |      4000 | ｜  |  3.9%   |  0.737  | 5  | ｜  |    6.4%     |   0.689   |    6 | ｜  |    14.4%    |  0.647  |    6 | ｜  |    38.5%    |  0.595  |    6     |
> >| **speechtokenizer** |       500 | ｜  | 15.2%|  0.334  |    9     | ｜  |  33.9% |   0.324   |    9     | ｜  |    64.2%    |  0.301  |    9     | ｜  |    91.7%    |  0.277  |    10    |
> >| **speechtokenizer** |      2000 | ｜  |    7.3%     |  0.773  |    8     | ｜  |  16.3%  |   0.734   |    8     | ｜  |    39.4%    |  0.682  |    8     | ｜  |    73.8%    |  0.600  |    8     |
> >| **speechtokenizer** |      4000 | ｜  |    4.4%     |  0.864  |    2     | ｜  |    8.1%     |   0.822   |    4     | ｜  |    21.0% |  0.765  |    4     | ｜  |    49.2%    |  0.684  |    5     |
> >| **BigCodec**        |      1040 | ｜  |    10.1%    |  0.829  |    7     | ｜  |    17.8%    |   0.785   |    7     | ｜  |    33.5% |  0.718  |    7     | ｜  |    63.0%    |  0.625  |    6     |
> >| **mimi**            |      1000 | ｜  |    5.1%     |  0.804  |    6     | ｜  |    7.8%     |   0.772   |    5     | ｜  |    14.7%    |  0.726  |    4     | ｜  |    33.6%    |  0.673  |    4     |
> >| **s3 token**        |       600 | ｜  |    3.9%     |  0.860  |    2     | ｜  |    5.2%     |   0.841   |    2     | ｜  |    8.1%     |  0.815  |    2     | ｜  |    16.7%    |  0.779  |    3     |
> >| **TASTE**           | **\~153** | ｜  |    4.8%     |  0.842  |  **4**   | ｜  |    5.3%     |   0.830   |  **3**   | ｜  |    6.9%     |  0.815  |  **2**   | ｜  |  **11.1%**  |  0.792  |  **1**   |
>
> > The results show that **our tokenizer remains stable across different noise levels, demonstrating strong robustness**.
>
> Follow up on: *[Q2] Can you report out-of-domain continuation ... on non-LibriVox corpora?*
> >We have conducted out-of-domain continuation on Expresso [1]. The result is presented below. Our TASTM consistently performs the best on the available metrics.
>
> > **Table R4-2**. Continuation experiment on Expresso (an out-of-domain dataset).
> > | Method | GPT-4o | UTMOS |
> > | :---- | :---: | :---: |
> > | TWIST 1.3B | 1.62 | 3.44 |
> > | TWIST 7B | 2.00 | 3.36 |
> > | Spirit LM | 2.24 | 3.34 |
> > | Spirit LM Expr. | 1.40 | 3.27 |
> > | TASLM 1B (embed.) | **3.21** | **3.65** |
>
> ### Reference
> [1] Nguyen, Tu Anh, et al. "Expresso: A benchmark and analysis of discrete expressive speech resynthesis." *Interspeech*, 2023.

---

### Official Review · Reviewer_tUyK · 2025-11-01

**Soundness:** 3
**Presentation:** 2
**Contribution:** 2
**Rating:** 4
**Confidence:** 3

**Summary:**

This paper introduces TASTE (Text-Aligned Speech Tokenization and Embedding), a method for learning text-aligned speech representations that directly bridge the modality gap between text and speech. The approach aggregates speech features using an attention-based mechanism guided by text transcriptions and is trained end-to-end with a speech reconstruction objective. This alignment produces compact and semantically consistent speech embeddings that preserve paralinguistic information while reducing sequence length. By adapting a pre-trained text language model with low-rank adaptation, the resulting joint model can process and generate speech coherently without additional alignment heuristics. Experiments show that TASTE improves speech continuation and next-speech prediction performance compared to existing spoken language models, demonstrating its effectiveness for unified text-speech modeling.

**Strengths:**

The paper tackles an important problem in spoken language modeling: how to represent speech and text within a shared embedding space for joint generation. The proposed method is conceptually simple yet effective, combining attention-based feature aggregation with an end-to-end reconstruction objective to achieve text-aligned speech embeddings. The learned representations capture both linguistic and paralinguistic information in a compact form and integrate smoothly with pre-trained text LMs through lightweight adaptation. The experimental results are consistent across multiple tasks, and the qualitative analyses illustrate interpretable alignment behavior. The paper is clearly written and provides a strong empirical demonstration of how modality alignment can improve spoken language modeling.

**Weaknesses:**

While the paper is technically sound, several aspects of the design and analysis remain underexplored. The attention-based aggregation is central to the method, yet its behavior and learned alignment patterns are not clearly analyzed, making it difficult to understand how alignment emerges during training. The paper lacks ablations isolating the effects of key design factors such as the chosen encoder layers, the reconstruction objective, and the LoRA adaptation on the joint model’s performance. Moreover, the evaluation focuses mainly on speech continuation and next-speech prediction, without examining general spoken understanding tasks or perceptual quality metrics that would better validate the claim of improved paralinguistic preservation. These gaps limit the interpretability and completeness of the study.

**Questions:**

- How exactly is the attention map used in the alignment process, and how sensitive is performance to layer selection for keys and values?
- What is the computational overhead of training the alignment module compared to standard speech tokenizers?
- Could the model be extended to use unpaired speech in a semi-supervised manner to improve robustness?
- How does TASTE handle inter-word silences or pauses during alignment and embedding generation? Are such regions ignored, or do they get represented implicitly in the embeddings?
- What is the rationale for using the last hidden layer as key and shallow hidden layers as value? Have other layer combinations been tested, and how sensitive is performance to this choice?

---

> ### Author Response · Authors · 2025-11-21
>
> We thank the reviewer for the thorough and constructive comments. Please find the response to your questions below.
>
> *[W1] The attention-based aggregation is central to the method, yet its behavior and learned alignment patterns are not clearly analyzed, making it difficult to understand how alignment emerges during training.*
>
> > To give more straightforward evidence that the aggregator has learned the alignment behavior, we draw the cross attention map of each head in our aggregator. The Figure can be found in our revision, Appendix A.1.2. The visualization clearly shows the aligned behavior on some of the heads, especially in the last layer (Figure 4). Note that the aligned behavior is automatically learned by the aggregator throughout the reconstruction. We do not provide any explicit alignment.
>
> *[W2] The paper lacks ablations isolating the effects of key design factors such as the chosen encoder layers, the reconstruction objective, and the LoRA adaptation on the joint model’s performance.*
>
> > We emphasize that we have conducted an ablation study on the design choice of using the shallow hidden in Table 4 of the main text. As discussed in Section 4.2.3 (5.2.3 in the revision), using the last hidden features as the cross attention value in the aggregator actually leads to lower S3 unit prediction accuracy. The results further strengthen our design choice of using shallow features empirically. As for the reconstruction objective and LoRA adaptation, we argue that these ablations are computationally expensive and we may not be able to include them during the rebuttal period. This does not hinder our main claim since the results have demonstrated the effectiveness of our tokenization for the SLM.
>
> *[W3] Moreover, the evaluation focuses mainly on speech continuation and next-speech prediction, without examining general spoken understanding tasks or perceptual quality metrics that would better validate the claim of improved paralinguistic preservation.*
>
> > The evaluation on pretrained SLMs follows typical standards as in the previous literature. And we have included human evaluation and UTMOS in Table 2. Moreover, we demonstrate that our pretrained SLM is capable of doing spoken question answering under the few-shot scenario, while all the other pretrained SLMs are not able to achieve. Turning the pretrained SLM into a model beyond that would require post-training, which is left for future work.
>
> *[Q1] How exactly is the attention map used in the alignment process, and how sensitive is performance to layer selection for keys and values?*
>
> > As mentioned in [W1], we visualize the cross attention map of each head in each layer in the revision. The results in Figure 4 clearly indicated the learned alignment behavior. As for the layer selection for keys and values, please refer to the discussion below in [Q5].
>
> *[Q2] What is the computational overhead of training the alignment module compared to standard speech tokenizers?*
>
> > Since our dataset is over 40,000 hours, it generally takes around 2 days on 8*A6000 to train the TASTE tokenizer. However, on a smaller subset (e.g., LibriTTS) would only take less than one day on an A6000 to finish the training, while still delivering reasonable reconstruction capability.
>
> *[Q3] Could the model be extended to use unpaired speech in a semi-supervised manner to improve robustness?*
>
> > Overall, we would describe our training scheme as weakly-supervised. We use Whisper as initialization of the tokenizer, and the dominant training dataset, Emilia, is also pseudo-labeled by Whisper. If we have unpaired speech data, we can perform pseudo-labeling as well to add more data into training to improve the robustness. We hope this answers your question to some extent.
>
> *[Q4] How does TASTE handle inter-word silences or pauses during alignment and embedding generation? Are such regions ignored, or do they get represented implicitly in the embeddings?*
>
> > This is a good question. We believe that they are modeled and represented implicitly in the embeddings. Our interpretation is based on the below reasons:
> > 1. Speech naturally includes a lot of pauses and inter-word silence, but Table 1 shows that our word-level duration consistency is high.
> > 2. We conduct visualization on the cross-attention of the aggregator, and we find out that some heads light up on the silence region. e.g., layer 0 head 19th.

---

> ### Author Response · Authors · 2025-11-21
>
> *[Q5] What is the rationale for using the last hidden layer as key and shallow hidden layers as value? Have other layer combinations been tested, and how sensitive is performance to this choice?*
>
> > We use the shallow hidden from the encoder since the target S3 units are also quantized from the shallow hidden layer of an ASR encoder. According to our ablation study in Table 4, directly using the shallow hidden leads to nearly 100% reconstruction accuracy (top-5); while using the last hidden is only 85%. This encourages us to use the shallow hidden as the aggregator's input as well. The aggregator is initialized from Whisper decoder, which already demonstrates word-level alignment capability from its cross-attention map of text query and last hidden as key. Combining these two ideas, we make the value obtained from shallow hidden, but keep using the text query and last hidden as key. The results in Table 4 shows that the performance of using this design is better than using the last hidden as both key and value, confirming our design choice. In general, using the last hidden works as well, but using the shallow hidden pushes the performance limit even further.

---

> > ### Comment · Reviewer_tUyK · 2025-11-23
> >
> > ## Comments from Reviewer
> >
> > Thank you for the detailed responses and the added visualizations. They provide helpful intuition about the alignment mechanism. However, I still have concerns regarding the design choice of using the **last encoder hidden layer as key** and the **shallow hidden layer as value**, which remain insufficiently explained.
> >
> > ### 1. Definition and selection of the shallow layer (ℓ)
> >
> > In the paper, ℓ is defined as a layer within `1 ≤ ℓ ≤ L/2`, but the revision and the visualizations (e.g., the first-layer example in Figure 4) suggest that a specific shallow layer is used in practice.
> >
> > It is not clear:
> > - **Which exact shallow layer** is used in the final model,
> > - Whether **different shallow layers were tested**, and
> > - How **sensitive performance is to the choice of ℓ**.
> >
> > Since the current ablation only compares “last hidden vs shallow hidden,” but not “which shallow layer,” it is difficult to judge the robustness of the design.
> >
> > ### 2. Justification for mixing key = h_L and value = h_ℓ
> >
> > Cross-attention usually assumes keys and values come from similar representational spaces. Here, `h_L` is a deep, semantic representation, and `h_ℓ` is an early, acoustic one.
> >
> > The response notes that Whisper decoder uses the last hidden layer as keys, but this does not clearly justify **why values can come from a different (shallow) layer**, nor why this discrepancy helps alignment. A more explicit explanation of why this combination is effective would be helpful.
> >
> > ### 3. Interpretation of the attention visualizations
> >
> > The added attention maps show aligned patterns, but it remains unclear how they support the specific key–value choice. For example:
> > - Would different shallow layers lead to different alignment behaviors?
> > - Does alignment degrade if both key and value come from the same layer?
> >
> > Without clarifying these connections, the visualizations do not fully justify the architectural decision.
> >
> > ---
> >
> > Clarifying the above points (particularly the **exact choice of ℓ**, the **sensitivity of performance to this choice**, and how the **attention maps validate the mixed-layer design**) would make the motivation and correctness of the proposed architecture much clearer.

---

> > > ### Author Response · Authors · 2025-11-25
> > >
> > > We thank the reviewer for the thoughtful and constructive comments. Please find our responses below.
> > >
> > > ---
> > >
> > > **Definition and selection of the shallow hidden layer ( $ \ell $ )**
> > >
> > > To be precise, our final model uses the 6-th encoder layer of Whisper as the shallow hidden representation. This choice is grounded in the characteristics of the S3 target units for two reasons:
> > >
> > > 1. S3 tokens are derived from a Whisper-style encoder.
> > >
> > > 2. In the original S3 tokenizer paper [1], the authors explicitly quantize the 6-th encoder layer of their Whisper-like model; these quantized codes are then used as S3 tokens.
> > >
> > > In short, S3 tokens originate from the 6-th layer of their ASR encoder. Our intuition is to select a hidden representation that is closest to the reconstructable target unit, which naturally points to shallow layers where more acoustic information is preserved. The definition
> > > $1 ≤ \ell ≤ L / 2$ was intended to formalize this intuition. However, we fully understand the reviewer’s concerns regarding (1) **how the shallow layer is chosen**, and (2) **how sensitive the method is to this choice**. We address both questions below.
> > >
> > > ---
> > >
> > > **About "how to choose the shallow hidden"**
> > >
> > > In the current work, we initially selected the 6-th layer based on the conjecture that it is closest to the S3 units, without yet providing an empirical justification or a systematic selection procedure.
> > > To improve completeness, we additionally conducted an empirical study measuring the similarity between each Whisper encoder layer and the S3 representations.
> > >
> > > We follow prior methodology [2] and apply Canonical Correlation Analysis (CCA), which identifies linear projections of two continuous random vectors that maximize their correlation [3]. This serves as a proxy for assessing how similar each encoder layer is to the target representation. The results are presented in Table R3-1.
> > >
> > >
> > > **Table R3-1**. The Canonical Correlation Analysis on each Whisper layer to S3 token embeddings.
> > > | Whisper Layer |  CCA w.r.t. S3  |
> > > |:-------------:|:-----:|
> > > | 1  | 0.80 |
> > > | 2  | 0.84 |
> > > | 3  | 0.85 |
> > > | 4  | **0.87** |
> > > | 5  | **0.88** |
> > > | 6  | **0.88** |
> > > | 7  | **0.88** |
> > > | 8  | **0.87** |
> > > | 9  | 0.86 |
> > > | 10 | 0.86 |
> > > | 11 | 0.85 |
> > > | 12 | 0.84 |
> > > | 13 | 0.83 |
> > > | 14 | 0.82 |
> > > | 15 | 0.82 |
> > > | 16 | 0.79 |
> > > | 17 | 0.77 |
> > > | 18 | 0.75 |
> > > | 19 | 0.73 |
> > > | 20 | 0.71 |
> > > | 21 | 0.67 |
> > > | 22 | 0.65 |
> > > | 23 | 0.63 |
> > > | 24 | 0.60 |
> > > | 25 | 0.57 |
> > > | 26 | 0.53 |
> > > | 27 | 0.52 |
> > > | 28 | 0.50 |
> > > | 29 | 0.47 |
> > > | 30 | 0.46 |
> > > | 31 | 0.41 |
> > > | 32 | 0.28 |
> > >
> > >
> > > As shown in Table R3-1, shallow layers--particularly layers 4 to 8--exhibit substantially higher correlations with the target representations. This provides strong support for our choice of
> > > $\ell=6$. Moreover, this analysis is lightweight and can be used as a preliminary procedure for selecting the shallow layer when switching to different target units. **We will include this discussion in the Appendix and we thank the reviewer for raising this concern.**
> > >
> > > ---
> > >
> > > **About "how sensitive it is to the selection"**
> > >
> > > Here, we would like to provide evidence that shows “the selection affects the performance **upper bound**, but is **not a limitation of our method**” below.
> > >
> > > 1. As shown in Table 4, even when using the last encoder layer as the value--which has the lowest correlation with the target units--the model still achieves substantially better performance than the text-only baseline. This demonstrates that the architecture remains effective even with suboptimal layer choices.
> > >
> > > 2. In [4], they show that the selection of the layers can be learned directly by the model. More specifically, in Section 6.2 of their paper, the leaned weights are 0.573 of the 8-th layer, 0.325 of the 16-th layer, 0.100 of the 24-th layer, and 0.0017 of the 32-th (last) layer (in average). Note that the learned weights also highlight the utilization of shallow hidden layers.
> > >
> > > Together, these observations suggest that although different choices of $\ell$ may affect the performance ceiling, the method is robust overall and does not rely critically on selecting a specific shallow layer.
> > >
> > > We thank the reviewer again for the prompt follow-up and for clearly outlining the remaining concerns. We hope the above responses address them adequately, and we would be happy to clarify further if needed.
> > >
> > > ---
> > >
> > > ### Reference
> > > [1] An, Keyu, et al. "Funaudiollm: Voice understanding and generation foundation models for natural interaction between humans and llms." *arXiv preprint* , 2024.
> > >
> > > [2] Pasad, Ankita, Ju-Chieh Chou, and Karen Livescu. "Layer-wise analysis of a self-supervised speech representation model." *ASRU*, 2021.
> > >
> > > [3] Hotelling, Harold. "Relations between two sets of variates." *Breakthroughs in statistics: methodology and distribution*, 1992.
> > >
> > > [4] Hsu, Ming-Hao, et al. "TASLA: Text-Aligned Speech Tokens with Multiple Layer-Aggregation." *arXiv preprint*, 2025.

---

### Official Review · Reviewer_8SAU · 2025-11-01

**Soundness:** 2
**Presentation:** 3
**Contribution:** 2
**Rating:** 4
**Confidence:** 5

**Summary:**

The paper presents TASTE, a text-aligned speech tokenization approach that uses transcription-guided cross-attention and RVQ to better align speech and text representations. The goal is to address the speech-text alignment mismatch common in spoken language modeling. While the motivation is clear and the design is coherent, the method largely builds upon existing codec frameworks with a text-conditioned aggregation step. Experimental results are competitive but not stronger than prior models such as SpeechTokenizer, Mimi, and EnCodec, and the evaluation omits more recent baselines. Overall, the contribution is conceptually sound but represents a modest advancement without clear empirical evidence of improved alignment and performance.

**Strengths:**

1. Using dynamic token frequency for better alignment with text is an interesting and well-motivated approach. It directly targets the speech-text mismatch that often limits speech tokenizer alignment.


2. Deriving speech tokens based on transcription-guided cross-attention is an interesting design choice that offers an interpretable mapping between continuous speech features and discrete text-aligned tokens.


3. The text-aligned editing demonstration is interesting. Swapping word-level TASTE tokens to transfer duration and other attributes in a controlled way is well-motivated and could inspire future work on speech style transfer or voice cloning.

**Weaknesses:**

1. The core idea of using text to guide lower-frequency speech tokens is sensible, but similar strategies have appeared before. Prior works have reduced speech token rates or trained tokenizers with auxiliary text objectives/distillation. The paper should better position TASTE relative to SpeechTokenizer, TWIST, DM-Codec, TadiCodec, and other text included tokenization approaches. It should clearly delineate what is new beyond the addition of a cross-attention aggregator and RVQ. The claim that TASTE is the “first end-to-end reconstruction objective to learn joint tokenization” is not fully substantiated because the distinction from prior work remains unclear.

2. The experimental evaluation is constrained. In Table 1, TASTE is compared only against SpeechTokenizer, Mimi, and EnCodec. While these are recognized baselines, they are relatively older and weaker compared to recent models such as WavTokenizer, DM-Codec, FACodec, and BigCodec, which have shown superior reconstruction performance. Similarly, Table 2 compares only to TWIST and SpiritLM, and should include newer SLMs approaches (Cosyvoice2, SparkTTS, and Llasa). Without such comparisons, the reported results lack convincing evidence of improvement.

3. TASTE does not outperform existing baselines (e.g., SpeechTokenizer, and EnCodec) across key metrics (WER, ViSQOL, Duration Consistency, Speaker Similarity, and MUSHRA). This raises questions about the practical benefit of the more complex and resource-intensive TASTE training pipeline. The paper repeatedly attributes its lower performance to “lower bitrate,” but this argument is unconvincing without matched-bitrate experiments. The authors should directly compare TASTE with low-bitrate variants of established codecs (e.g., DAC, WavTokenizer, XCodec2, BigCodec). Without such controlled comparisons, the main claim of improved speech tokenization remains weak.

4. The design choice of setting the aggregator's keys as last-layer encoder features (K = h(L)) and values as shallow-layer features (V = h(l)) is based on questionable representational assumption and not supported with experiments. The paper assumes h(L) captures alignment cues while h(l) preserves acoustic detail, but offers no empirical analysis (e.g., attention visualizations or layer ablations) to support this. It is unclear whether the observed effects are from meaningful alignment learning or coincidental correlations.

5. The ASR dependence of TASTE limits general applicability. The model requires accurate transcriptions to derive aligned tokens, yet there is no analysis of robustness under realistic noisy ASR or low-resource conditions. Without such tests, the practical viability of TASTE is uncertain. Moreover, The evaluation is also restricted to English corpora (LibriSpeech, Emilia). This ASR dependence also limit TASTE's applicability to spontaneous or multilingual speech. Given that TASTE explicitly depends on text-speech alignment, it is unclear whether it generalizes to languages with more complex non-phonemic orthography.

**Questions:**

1. Please see the Weakness section for points where additional analysis, clarification, or experimentation would strengthen the paper.
2. Can the authors provide clearer evidence that the observed improvements come from genuine text-speech alignment rather than architectural or bitrate differences? Specifically, can they (i) compare with stronger recent baselines under matched bitrates, (ii) justify the representational choice of K = h(L), V = h(l) with empirical analysis, and (iii) evaluate robustness under noisy, multilingual or imperfect ASR to confirm practical applicability?

---

> ### Author Response · Authors · 2025-11-21
>
> We thank the reviewer for the thorough and constructive comments. Please find the response to your questions below.
>
> *[W1] The core idea of using text to guide lower-frequency speech tokens is sensible, but similar strategies have appeared before.*
>
> > We want to clarify we are the first one to build a text-aligned (dynamic in frame rate) joint tokenization for joint SLMs. Our joint tokenization **completely mitigates the length mismatch** problem while demonstrating a good reconstruction quality. We are the first to propose and build up this joint tokenization **specialized for SLM**, and we further demonstrate the effectiveness of it with the pretrained SLM experiments.
>
> *...The claim that TASTE is the “first end-to-end reconstruction objective to learn joint tokenization” is not fully substantiated because the distinction from prior work remains unclear.*
>
> > Thank you for pointing this out. Our claim was not that TASTE is the "first end-to-end reconstruction objective to learn tokenization" in general, but specifically what we stated at the end of the abstract: "the first end-to-end approach that utilizes a reconstruction objective to learn a joint tokenization and embedding **tailored for text–speech spoken language modeling**." Prior work may use reconstruction to learn either speech-only units, but they do not learn a joint, text-aligned speech tokenization together with its embedding for unified text–speech generation. Our contribution lies in this joint text–speech formulation that directly mitigates the length mismatch during tokenization, not in being the first to use reconstruction for tokenization.
>
> *[W2] The experimental evaluation is constrained. In Table 1, TASTE is compared only against SpeechTokenizer, Mimi, and EnCodec. While these are recognized baselines, they are relatively older and weaker compared to recent models such as WavTokenizer, DM-Codec, FACodec, and BigCodec, which have shown superior reconstruction performance.*
>
> > Thank you for raising this concern. To address this, we try our best to include more baselines in the rebuttal period. The results are presented below in **Table R2-1**.
> >
> > **Table R2-1**. Speech reconstruction results with more baselines. Our method surpasses all the additional baselines under 1000 bitrate by a huge margin, demonstrating its effectiveness.
> >| Methods        | Freq. | bitrate | ASR-WER (↓) | UTMOS | DNS-MOS |
> >| -------------- |:-----:|:-------:|:-----------:|:-----:|:-------:|
> >| DAC         |  50   |   500   |    74.9%    | 1.25  |  2.54   |
> >| DAC         |  50   |  1000   |    13.3%    | 1.29  |  2.90   |
> >| DAC          |  50   |  12000  |    2.2%     | 4.00  |  3.78   |
> >| DM-Codec     |  50   |   500   |    69.8%    | 1.50  |  2.94   |
> >| DM-Codec     |  50   |  1000   |    10.3%    | 2.42  |  3.11   |
> >| DM-Codec     |  50   |  8000   |    2.5%     | 3.46  |  3.34   |
> >| BigCodec       |  80   |  1040   |    3.0%     | 4.11  |  3.86   |
> >| WavTokenizer   |  40   |   480   |    11.3%    | 3.57  |  3.78  |
> >| TASTE          |  ~3   |  ~150   |    4.4%     | 4.29  |  4.10   |
>
> *...Similarly, Table 2 compares only to TWIST and SpiritLM, and should include newer SLMs approaches (Cosyvoice2, SparkTTS, and Llasa)*
>
> > We would like to emphasize that our SLM is a pretrained spoken language model, aiming to build an universal speech processing system, following the definition in [1]. We compare our method with the available pretrained SLMs and have included our own baseline (jointly modeling on S3 tokens). The evaluation typically follows the previous work. The above-mentioned models (CosyVoice, SparkTTS, Llasa) are all task-specific, autoregressive-based TTS models, which are incomparable with our pretrained SLM.
>
> *[W3] TASTE does not outperform existing baselines (e.g., SpeechTokenizer, and EnCodec) across key metrics... The paper repeatedly attributes its lower performance to “lower bitrate,” but this argument is unconvincing without matched-bitrate experiments.*
>
> > Our TASTE tokenization is specialized for joint SLM. It is reasonable that its performance on speech reconstruction is inferior to tokenization with thousands of bps. However, comparing with the most “matched-bitrate” scenario, it is obvious that TASTE is much better. This can be evidenced by Table 1 of the paper and the additional results in Table R2-1, in which we include most of the tokenization methods that the reviewer mentioned.
>
> *...Without such controlled comparisons, the main claim of improved speech tokenization remains weak.*
>
> > We clarify that we are not aiming to build a perfect speech tokenizer for reconstruction but an effective one for joint SLM. We conduct speech reconstruction just to demonstrate that our text-aligned speech tokenization carries rich paralinguistic information despite its low bitrate. The more important thing is that TASTE, a joint tokenization tailored for joint SLM, could actually lead to an effective joint SLM, as shown by Table 2 and 3.

---

> ### Author Response · Authors · 2025-11-21
>
> *[W4] The design choice of setting the aggregator's keys as last-layer encoder features (K = h(L)) and values as shallow-layer features (V = h(l)) is based on questionable representational assumption and not supported with experiments.*
>
> > We emphasize that we have conducted an ablation study on the design choice of using the shallow hidden in Table 4 of the main text. As discussed in Section 4.2.3 (5.2.3 in the revision), using the last hidden features as the cross attention value in the aggregator actually leads to lower S3 unit prediction accuracy. The results further strengthen our design choice of using shallow features empirically.
>
> *...but offers no empirical analysis (e.g., attention visualizations or layer ablations) to support this...*
>
> > In addition, to visualize the learned alignment behavior, we draw the cross attention map of each head in our aggregator. Figure 4, 5 in Appendix A.1.2 illustrate this and can be found in our revision. The visualization clearly shows the aligned behavior on some of the heads, especially in the last layer (Figure 4). Note that the aligned behavior is automatically learned by the aggregator throughout the reconstruction. We do not provide any explicit alignment.
>
> *[W5] The ASR dependence of TASTE limits general applicability. The model requires accurate transcriptions to derive aligned tokens, yet there is no analysis of robustness under realistic noisy ASR or low-resource conditions. Moreover, The evaluation is also restricted to English corpora (LibriSpeech, Emilia).*
>
> > We agree that TASTE requires an ASR. Recent advances in robust and efficient ASR systems [2] provide sufficiently reliable transcripts for our setting, enabling us to focus on the core goal of deriving high-quality text-aligned speech tokens. While evaluating robustness under noisy or low-resource ASR conditions is indeed valuable, such analyses are orthogonal to the contributions we aim to establish in this work. Our experimental setup for both the tokenizer and SLM follows standard practice in prior work, and we have conducted ablation regarding ASR error propagation in Appendix A.2.1. Extending TASTE to more challenging ASR conditions and to other languages is a natural direction for future work. To further strengthen the tokenization robustness, we are conducting controlled experiments on different noise levels, and will put up the results once we are done **(completed, please see the next official comment)**.
>
> *[Q1] Please see the Weakness section for points where additional analysis, clarification, or experimentation would strengthen the paper.*
>
> > Please see the above.
>
> *[Q2] ...can they (i) compare with stronger recent baselines under matched bitrates, (ii) justify the representational choice of K = h(L), V = h(l) with empirical analysis, and (iii) evaluate robustness under noisy, multilingual or imperfect ASR to confirm practical applicability?*
>
> > (i): Please see [W2].
> > (ii): Please see [W4].
> > (iii): Please see [W5].
>
> ### References
> [1] Arora, Siddhant, et al. "On the landscape of spoken language models: A comprehensive survey." *TMLR*, 2025.
>
> [2] Radford, A., et al. "Robust speech recognition via large-scale weak supervision (arXiv: 2212.04356). *ICML*, 2023.

---

> ### Author Response · Authors · 2025-12-02
>
> > We have finished the robustness ablation study of our tokenizer to address the reviewer's concern (*[W5]* in the previous official comment). The results are presented below.
>
> > **Table R2-2**. The robusness ablatioin study of our tokenization. We introduce different levels of white noise to evaluate how our tokenizer performs under varying noise conditions. Here, SNR denotes the signal-to-noise ratio, ranging from 20 dB (nearly clean) to 5 dB (very noisy). We report two metrics: ASR-WER as an indicator of reconstruction quality, and speaker similarity (SIM) as a measure of reconstruction similarity. For ease of comparison, we also provide the overall rank of each tokenizer considering both metrics.
> >
> >|                     |           | ｜  |  SNR=20dB   | (almost |  clean)  | ｜  |  SNR=15dB   | (slightly |  noisy)  | ｜  |  SNR=10dB   | (fairly |  noisy)  | ｜  |   SNR=5dB   |  (very  |  noisy)  |
> >|:------------------- | ---------:|:---:|:-----------:|:-------:|:--------:|:---:|:-----------:|:---------:|:--------:|:---:|:-----------:|:-------:|:--------:|:---:|:-----------:|:-------:|:--------:|
> >| **Method**          |   **bps** | ｜  | **ASR-WER** | **SIM** | **Rank** | ｜  | **ASR-WER** |  **SIM**  | **Rank** | ｜  | **ASR-WER** | **SIM** | **Rank** | ｜  | **ASR-WER** | **SIM** | **Rank** |
> >| **groud-truth**     |      256k | ｜  |    2.3%     |   \-    |          | ｜  |    2.5%     |    \-     |          | ｜  |    3.6%     |   \-    |          | ｜  |    9.2%     |   \-    |          |
> >| **DAC**             |       500 | ｜  |    93.7%    |  0.193  |    13    | ｜  |    97.7%    |   0.201   |    12    | ｜  |    98.6%    |  0.205  |    12    | ｜  |    98.3%    |  0.202  |    13    |
> >| **DAC**             |      1000 | ｜  |    36.0%    |  0.292  |    11    | ｜  |    55.0%    |   0.276   |    11    | ｜  |    80.2%    |  0.278  |    11    | ｜  |    94.5%    |  0.283  |    10    |
> >| **DAC**             |     12000 | ｜  |    2.5%     |  0.937  |    1     | ｜  |    2.9%     |   0.924   |    1     | ｜  |    4.7%     |  0.914  |    1     | ｜  |    12.3%    |  0.907  |    1     |
> >| **DM-Codec**        |       500 | ｜  |    98.7%    |  0.295  |    12    | ｜  |   104.6%    |   0.264   |    12    | ｜  |   107.1%    |  0.264  |    12    | ｜  |   105.8%    |  0.283  |    12    |
> >| **DM-Codec**        |      1000 | ｜  |    33.9%    |  0.479  |    9     | ｜  |    53.0%    |   0.447   |    9     | ｜  |    77.4%    |  0.435  |    9     | ｜  |    97.1%    |  0.415  |    9     |
> >| **DM-Codec**        |      4000 | ｜  |    3.9%     |  0.737  |    5     | ｜  |    6.4%     |   0.689   |    6     | ｜  |    14.4%    |  0.647  |    6     | ｜  |    38.5%    |  0.595  |    6     |
> >| **speechtokenizer** |       500 | ｜  |    15.2%    |  0.334  |    9     | ｜  |    33.9%    |   0.324   |    9     | ｜  |    64.2%    |  0.301  |    9     | ｜  |    91.7%    |  0.277  |    10    |
> >| **speechtokenizer** |      2000 | ｜  |    7.3%     |  0.773  |    8     | ｜  |    16.3%    |   0.734   |    8     | ｜  |    39.4%    |  0.682  |    8     | ｜  |    73.8%    |  0.600  |    8     |
> >| **speechtokenizer** |      4000 | ｜  |    4.4%     |  0.864  |    2     | ｜  |    8.1%     |   0.822   |    4     | ｜  |    21.0%    |  0.765  |    4     | ｜  |    49.2%    |  0.684  |    5     |
> >| **BigCodec**        |      1040 | ｜  |    10.1%    |  0.829  |    7     | ｜  |    17.8%    |   0.785   |    7     | ｜  |    33.5%    |  0.718  |    7     | ｜  |    63.0%    |  0.625  |    6     |
> >| **mimi**            |      1000 | ｜  |    5.1%     |  0.804  |    6     | ｜  |    7.8%     |   0.772   |    5     | ｜  |    14.7%    |  0.726  |    4     | ｜  |    33.6%    |  0.673  |    4     |
> >| **s3 token**        |       600 | ｜  |    3.9%     |  0.860  |    2     | ｜  |    5.2%     |   0.841   |    2     | ｜  |    8.1%     |  0.815  |    2     | ｜  |    16.7%    |  0.779  |    3     |
> >| **TASTE**           | **\~153** | ｜  |    4.8%     |  0.842  |  **4**   | ｜  |    5.3%     |   0.830   |  **3**   | ｜  |    6.9%     |  0.815  |  **2**   | ｜  |  **11.1%**  |  0.792  |  **1**   |
>
> > The results show that **our tokenizer remains stable across different noise levels, demonstrating strong robustness**. Notably, TASTE achieves the best ASR-WER under the noisiest condition. This suggests that the underlying ASR system is not a limiting factor for the general applicability of TASTE; rather, it serves as an effective and reliable source of semantic information for the reconstruction.

---

### Official Review · Reviewer_wxPH · 2025-11-09

**Soundness:** 2
**Presentation:** 3
**Contribution:** 3
**Rating:** 6
**Confidence:** 3

**Summary:**

The paper proposes a novel tokenizer, called TASTE, specifically designed for text audio generative modeling. The motivation is that speech tokens generally have much longer length for the same content compared to their text counterpart. The proposed approach generates speech tokens of equal length as textual tokens. Technically, this is achieved by first passing the audio through a pre-trained speech encoder (Whisper). From this speech encoder, both last layer and shallow hidden states are extracted and fed into an attention based aggregator as key and value respectively. The queries for this aggregator are initialized by a text transcription of the speech. Since the queries are based on text, the final speech tokens match in length to the text tokens. The paper provides an intuition that these speech tokens focus on learning paralinguistic information as they are supposed to be used along with the corresponding text tokens. For decoding, a speech decoder uses both the text and speech tokens and decodes them back to speech using a vocoder. The tokenizer is trained on a speech resynthesis task wiht a cross entropy loss to predict the speech unit and commitment losses on the residual codebooks.

Apart from the tokenizer, the paper also proposes a speech language model which integrates the TASTE tokenizer, called TASLM. There are two variants - one which trains on the discrete codebook indices similar to a traditional language model and one which directly operates on the continuous latent embeddings.

Experiments are done by training on Emilia and LibriTTS datasets and evaluating on LibriTTS test split. The paper evaluates the tokenizer for speech reconstruction, speech editing and  TASLM on speech continuation, likelihood-based next-speech selection and spoken question answering.

**Strengths:**

- The paper proposes an interesting way to formulate tokenizers by using multimodal speech text tokens to provide complimentary information. This can help in significantly reducing the number of tokens needed to model speech.
- It is very interesting to see the behavior of this model in text-aligned speech editing. The model seems to be able to take paralingual information from one speech sequence and transfer it to another containing the same content. It would be interesting to pursue on this further and understand whether this type of editing can be done across very different voice types, like belonging to different genders etc.

**Weaknesses:**

- The proposed TASTE tokenizer tends to learn text aligned speech embeddings. One limitation of such an approach would be for sounds that occur commonly in speech but may not have a textual counterpart. For example, if a person laughs, it is unclear how the corresponding textual token would be retrieved. The approach may _collapse_ the laughing related information to the neighboring speech tokens if they have textual counterparts.
- The paper does not evaluate on the sWUGGGY and sBLIMP ZeroSpeech 2021 benchmark which is standard in literature [1,2]. These benchmark test the lexical and grammatical knowledge of the model to understand how well model can identify real words vs phoenetically similar non-words. I believe this might be a limitation of this tokenization approach as it needs a text transcription of the speech input.
- The paper does not explain its underperforming benchmarks. For example, In table 1, the proposed method has significantly worse WER compared to other tokenizers but the paper does not dive deeper into it. Instead, the text focuses on how WER is better than text-only baseline (343-344). Similarly, for table 2, it would be good to understand what factors contribute to the underperformance on SALMON benchmark. In table 8, this underperformance seems to happen for most of the experiments in SALMON and some discussion around it would be helpful.
- While the paper cites [2], it is unclear why it is not compared against the experiments of the paper.
- Related work section is missing from the main paper. While it is present in the appendix, it should be a part of the main paper.

References:
1. Spirit LM: Interleaved Spoken and Written Language Model, ACL 2025
2. Align-SLM: Textless Spoken Language Models with Reinforcement Learning from AI Feedback, ACL 2025

**Questions:**

- As the paper emphasizes that the goal for the speech tokens is to learn paralinguistic information, is there any sort of text encoding done to learn semantic information from the language?
- It is unclear to me how the decoding is done from TASLM output. Since the decoder requires both text and speech, how is it achieved?

**Details Of Ethics Concerns:**

I am not sure if this would require an ethics review, but the paper uses human evaluators in the MUSHRA benchmark. The paper mentions they use Amazon Mechanical Turk getting human evaluators. It would be good to understand whether annotators were fairly compensated.

---

> ### Author Response · Authors · 2025-11-21
>
> We thank the reviewer for the thorough and constructive comments. Please find the response to your questions below.
>
> *[W1] ...One limitation of such an approach would be for sounds that occur commonly in speech but may not have a textual counterpart.*
>
> > Thank you for raising this point. We acknowledge that this is a limitation for current work, as discussed in Limitation. To address this, possible solutions could be **1)** using an audio captioner or **2)** introducing non-verbal event tags. These are all under-explored ideas for us to create a more general audio  joint tokenization for joint SLM.
>
> *[W2] The paper does not evaluate on the sWUGGGY and sBLIMP ZeroSpeech 2021 benchmark which is standard in literature (SpiritLM, Aligned-SLM).*
>
> > We do not evaluate our SLM on the sWUGGY and sBLIMP ZeroSpeech 2021 benchmark due to the below reasons:
> > 1. Their evaluation focuses on semantic plausibility, which overlaps with the StoryCloze benchmark.
> > 2. The use of text from ASR is often considered as topline (SpiritLM, Aligned-SLM).
> >
> > We understand that the use of ASR text in the tokenization stage may introduce some limitations (as discussed in the previous point). However, we would like to gently clarify that this is not the main reason why we do not evaluate the sWUGGY and sBLIMP.
>
> *[W3] The paper does not explain its underperforming benchmarks.*
>
> >In Table 1, we emphasize that we are not aiming to build a perfect speech tokenizer but an effective one for joint SLM. On ASR-WER, we demonstrate that our tokenization is effective (with the lowest bitrate and frequency, while being comparable to other higher-bitrate tokenizations). To strengthen this, we also include results from more recent methods, presented as follows:
>
> > **Table R1-1**. Speech reconstruction results with more baselines. Our method surpasses all the additional baselines under 1000 bitrate by a huge margin, demonstrating its effectiveness.
> >| Methods        | Freq. | bitrate | ASR-WER (↓) | UTMOS | DNS-MOS |
> >| -------------- |:-----:|:-------:|:-----------:|:-----:|:-------:|
> >| DAC            |  50   |   500   |    74.9%    | 1.25  |  2.54   |
> >| DAC            |  50   |  1000   |    13.3%    | 1.29  |  2.90   |
> >| DAC            |  50   |  12000  |    2.2%     | 4.00  |  3.78   |
> >| DM-Codec       |  50   |   500   |    69.8%    | 1.50  |  2.94   |
> >| DM-Codec       |  50   |  1000   |    10.3%    | 2.42  |  3.11   |
> >| DM-Codec       |  50   |  8000   |    2.5%     | 3.46  |  3.34   |
> >| BigCodec       |  80   |  1040   |    3.0%     | 4.11  |  3.86   |
> >| WavTokenizer   |  40   |   480   |    11.3%    | 3.57  |  3.78   |
> >| TASTE          |  ~3   |  ~150   |    4.4%     | 4.29  |  4.10   |
>
> *...it would be good to understand what factors contribute to the underperformance on SALMON benchmark.*
>
> > We observe that our TASLM falls short for background-related attributes (Room, Background) where the samples are added with environmental sounds (echoes, instruments, background noises from FSD50K). Since TASTE tokenization focuses on natural speech and has not trained on audio with environmental sound and noise, it may fall short to convey such information. However, on the speech related attributes, such as gender and speaker, our TASLM performance is much better and is comparable to other SLMs. We have included such discussion in Appendix A.1.3 in the revision.
>
> *[W4] While the paper cites [Align-SLM], it is unclear why it is not compared against the experiments of the paper.*
>
> > Thanks for raising this point. We can not compare our method with Align-SLM simply because the model and codebase are unavailable. We can’t get its speech continuation results.
>
> *[W5] Related work section is missing from the main paper. While it is present in the appendix, it should be a part of the main paper.*
>
> > Thank you for the suggestion. We have updated our manuscript to include the section as the space has extended, and it is reflected in our revision.

---

> ### Author Response · Authors · 2025-11-21
>
> *[Q1] As the paper emphasizes that the goal for the speech tokens is to learn paralinguistic information, is there any sort of text encoding done to learn semantic information from the language?*
>
> > We are not quite sure about the meaning of “text encoding”. In practice, we use an ASR (Whisper) to extract the text transcript. Then the text transcript is tokenized by a conventional BPE tokenizer. Please let us know if you have further questions about this point.
>
> *[Q2] It is unclear to me how the decoding is done from TASLM output. Since the decoder requires both text and speech, how is it achieved?*
>
> > We employ a multi-head prediction scheme of our SLM that simultaneously predicts text and speech tokens. Then, the speech decoder takes the output tokens from the TASLM as input conditions and then generates the speech unit (S3 unit) in an auto-regressive manner, followed by a unit-to-speech vocoder to generate the final speech waveform.
>
> *[E] I am not sure if this would require an ethics review, but the paper uses human evaluators in the MUSHRA benchmark. The paper mentions they use Amazon Mechanical Turk getting human evaluators. It would be good to understand whether annotators were fairly compensated.*
>
> > Thank you for pointing this out. We have included an ethical statement in our revision to address the above concern.

---

### Author Response · Authors · 2025-12-03
**Author Final Remarks**

Dear Area Chairs and Reviewers,

We sincerely thank the reviewers for the thoughtful evaluations and constructive discussions. Below, we summarize the key contributions of our work and how the discussion addressed the main concerns raised during the review process.

---

### **Novelty and Contributions Summary**

Given the recent shift from pure spoken language models (SLMs) towards hybrid joint SLMs, our work is, to the best of our knowledge:

1. The first to consider joint tokenization specifically for joint SLMs.

2. A joint tokenization method that is both simple and effective.

3. Demonstrating significant advantages over conventional pure speech tokenization for joint SLMs.

All the reviewers have recognized our main contributions: **1) mitigating the length mismatch problem in a clean way druing tokenization** which **2) efficiently lead to building an effective joint SLM**.

---

### **Author-Reviewer Discussion Summary**

Below, we summarize the key concerns raised by the reviewers and the corresponding evidence we provide to address each of them.

1. The robustness of TASTE tokenization
Several reviewers asked whether the joint tokenizer remains reliable under noise. In response, we added robustness ablation study to verify the effectiveness of our tokenizer across multiple SNR levels. The results show that **TASTE maintains stable performance and is substantially more robust than prior methods [1]**.

2. Request for more recent baselines
Reviewer 8SAU requested us to conduct more recent baselines. We have conducted extensive experiment that covers almost all the methods Reviewer 8SAU mentioned. The results indicate that **TASTE consistently outperforms recent tokenizers under comparable bitrate setups [2]**, in line with the trends observed in Table 1.

3. Selection of the shallow hidden layer
Reviewer tUyK asked about the rationale for selecting the shallow hidden representation an how exactly the selection is. In addition to the textual explanation, we added **the cross attention visualization which clearly shows the speech-text alignment [3]**; and **an analysis that supports our selection of the shallow hidden layer [4]**.

All other reviewer concerns were addressed in detail in the individual rebuttals, and **the revised manuscript already reflects the major requested changes**, as listed below.

---

### **Revision Summary**

**[1]** Add robustness ablation study (Appendix A.1.4 + Table 8).

**[2]** Include more baselines for tokenization (Appendix A.1.4 + Table 8; Table R2-1 below).

**[3]** Add aggregator cross attention visualization (Appendix A.1.2 + Figure 4&5).

**[4]** Add discussion and analysis on shallow hidden selection (Appendix A.1.3 + Figure 6).

[5] Move related work from appendix to the main text (Section 2).

[6] Discuss the weighted-sum mechanism (Appendix A.5).

[7] Improve our ethics statements (before References).

---

We sincerely appreciate the reviewers’ constructive feedback, which has meaningfully improved the clarity, quality, and the completeness of our work. Thank you again for your time and thoughtful evaluation.

---

### Meta-Review · Area_Chair_togi · 2025-12-15

**Summary:**

This paper introduces Text-Aligned Speech Tokenization and Embedding (TASTE), a method that directly addresses the modality gap by aligning speech token with the corresponding text transcription during the tokenization stage. Across all five reviews, TASTE is seen as an interesting and practically useful approach for text-aligned speech tokenization that enables shorter speech sequences, better speech–text integration, and easier adaptation of LLMs into SLMs. The reviewers also describe the contribution as moderately novel, under-evaluated, and missing several critical comparisons and analyses. The authors did a good job on the rebuttal and addressed most of concerns.

**Reviewer Concerns:**

I think most of concerns have been addressed by the rebuttal.

**Reviewer Scores:**

I think the reviewer would have changed their scores.

---

### Decision · Program_Chairs · 2026-01-26

Accept (Poster)